# VALUE-BIASED MAXIMUM LIKELIHOOD ESTIMATION FOR MODEL-BASED REINFORCEMENT LEARNING IN DISCOUNTED LINEAR MDPS

## ABSTRACT

We consider the infinite-horizon linear mixture Markov Decision Processes (MDPs), where the transition probabilities of the dynamic model can be linearly parameterized with the help of a predefined low-dimensional feature mapping. While the existing regression-based approaches have been theoretically shown to achieve nearly-optimal regret, they are computationally rather inefficient due to the need for a large number of optimization runs in each time step, especially when the state and action spaces are large. To address this issue, we propose to solve linear mixture MDPs through the lens of Value-Biased Maximum Likelihood Estimation (VBMLE), which is a classic model-based exploration principle in the adaptive control literature for resolving the well-known closed-loop identification problem of Maximum Likelihood Estimation. We formally show that (i) VBMLE enjoys $\widetilde{O}(d\sqrt{T})$ regret, where $T$ is the time horizon and $d$ is the dimension of the model parameter, and (ii) VBMLE is computationally more efficient as it only requires solving one optimization problem in each time step. In our regret analysis, we offer a generic convergence result of MLE in linear mixture MDPs through a novel supermartingale construct and uncover an interesting connection between linear mixture MDPs and online learning, which could be of independent interest. Finally, the simulation results show that VBMLE significantly outperforms the benchmark methods in both empirical regret and computation time.

## 1 INTRODUCTION

Model-based reinforcement learning (MBRL) is one fundamental paradigm that learns an optimal policy by alternating between two subroutines: estimation of the transition dynamics and planning according to the learned dynamics model. MBRL has been extensively studied in the tabular setting from various perspectives, including (Auer et al., 2008; Azar et al., 2017), which have been shown to achieve either optimal regret bounds or sample complexity. Despite the above success, the conventional tabular MBRL methods are known to be computationally intractable in RL problems with large state or action spaces due to the need for direct estimation and access to the per-state transition probability. To enable MBRL for large state and action spaces, one important recent attempt is to study Markov decision processes with linear feature mappings (Zhou et al., 2021b), which is termed linear mixture MDP subsequently in this paper. Specifically, linear mixture MDPs assume that the probability of each transition can be represented by $\langle \phi(s'|s, a), \boldsymbol{\theta}^* \rangle$, where $\phi(\cdot|\cdot, \cdot)$ is a known feature function for each possible transition, and $\boldsymbol{\theta}^*$ parametrizes the transition probabilities to be learned. This framework can readily encompass various related formulations, such as tabular MDPs, feature-based linear transition models (Yang & Wang, 2019), the linear combination of base models (Modi et al., 2020), and linear value function frameworks (Zanette et al., 2020).

Based on the existing literature of linear mixture MDPs, the existing approaches could be divided into two primary categories depending on the length of the learning horizon: *episodic MDPs* and *infinite-horizon discounted MDPs*. In episodic MDPs, one important feature is that the environment state could be conveniently reset to some initial state when a new episode starts. Several recent works have explored episodic MDPs through the use of value-targeted regression techniques, e.g., (Ayoub et al., 2020; Zhou et al., 2021a). A more detailed survey of the related works for episodic MDPs is deferred to Section 2. By contrast, in infinite-horizon linear mixture MDPs, due to the

absence of the periodic restart, addressing exploration and conducting regret analysis could be even more challenging than that in episodic MDPs. Some recent attempts tackles infinite-horizon linear mixture MDPs by designing regression-based approaches and establishing theoretical guarantees (Zhou et al., 2021a;b; Chen et al., 2022). However, their algorithms could suffer from high computational complexity and be intractable in practice for the following reasons: (i) In the existing regression-based approaches, in each time step, one would need to solve a constrained optimization problem of the action-value function for *each* state-action pair. (ii) Moreover, in order to represent the value function as a linear combination of the learned parameter vector $\boldsymbol{\theta}$, it is necessary for those regression-based approaches to construct a vector $\phi_V(s, a) := \sum_{s' \in \mathcal{S}} \phi(s'|s, a) V(s')$. As a result, the action-value function can be expressed as follows: $Q(s, a) = \langle \phi_V(s, a), \boldsymbol{\theta} \rangle$. However, constructing $\phi_V$ could be computationally intractable when dealing with a large state space. These limitations render the regression-based approaches mentioned above rather challenging to implement and deploy in practice. Therefore, one important research question remains to be answered: *How to design an efficient model-based RL algorithm for infinite-horizon discounted linear mixture MDPs with provable regret guarantees?*

In this paper, we answer the above question affirmatively. Specifically, to address the above limitations, we design a tractable approach based on the classic principle of Value-Biased Maximum Likelihood Estimation (VBMLE) (Kumar & Lin, 1982), which has shown promising results in recent developments in bandits (Hung et al., 2021; Hung & Hsieh, 2023) and tabular RL (Mete et al., 2021), and leverage the value biasing technique to enforce exploration. The major advantage of VBMLE is that with the help of value biasing, it requires solving only one optimization problem for learning the dynamics model parameter at each time step and thereby enjoys a significantly lower computational complexity than the regression-based approaches. Moreover, we formally establish $\tilde{\mathcal{O}}(d\sqrt{T})$ regret bound based on the following novel insights: (i) We establish a convergence result on the Maximum Likelihood Estimator for linear mixture MDPs by using a novel supermartingale approach. (ii) Through this construct, we also find useful connections between (i) the linear mixture MDPs and online portfolio selection problem as well as (ii) VBMLE and the Follow-the-Leader algorithm in online learning. We highlight the main contributions as follows:

- We adapt the classic VBMLE principle to the task of learning the dynamic model for linear mixture MDPs. Our proposed algorithm addresses model-based RL for linear mixture MDPs from a distributional perspective, which learns the parameterized transition directly by maximum likelihood estimation without resorting to regression, and guides the exploration via value biasing instead of using concentration inequalities.

- We establish the theoretical regret bound of VBMLE by providing a novel theorem connected to the confidence ellipsoid of MLE. Furthermore, we uncover an interesting connection between online learning and our regret analysis.

- We conduct an empirical analysis to assess both the computational complexity and empirical regret performance. The simulation results demonstrate that VBMLE exhibits a clear advantage in terms of both effectiveness in regret and computational efficiency.

## 2 RELATED WORKS

**VBMLE for Multi-Armed Bandits and RL**. Regarding VBMLE, various prior works have applied this method to different bandit settings and tabular MDP. Firstly, (Liu et al., 2020) focuses on solving non-contextual bandits with exponential family reward distributions. Next, (Hung et al., 2021) introduces two variations of VBMLE: LinRBMLE and GLM-RBMLE. These methods are designed for solving linear contextual bandits and result in an index policy. Furthermore, (Hung & Hsieh, 2023) leverages the representation power of neural networks and proposes NeuralRBMLE. This approach is specifically designed for solving neural bandits, making no assumptions about the unknown reward distribution. As for the MDP setting, (Mete et al., 2021) has adapted VBMLE to solve tabular MDPs, where the states and actions belong to a known finite set, while (Mete et al., 2022) analyzed the finite performance of a constrained version of VBMLE. By contrast, this paper takes the very first step towards understanding the theoretical regret performance of VBMLE in RL beyond the tabular settings.

**Episodic MDPs with Function Approximation**. Based on the class of transition dynamics models, the episodic MDPs with function approximation could be divided into the following categories:

- *Linear mixture MDPs*: To tackle large MDPs, one common approach is to leverage linear mixture MDPs, which enables a compact model representation through feature mapping. For instance, from a model-free perspective, (Cai et al., 2020) proposes an optimistic variant of Proximal Policy Optimization algorithm (OPPO) to address exploration in linear mixture MDPs. (Ayoub et al., 2020) addresses linear mixture MDPs by proposing UCRL-VTR, which extends the classic UCRL algorithm (Jaksch et al., 2010) by using the value-targeted model regression as an optimistic approach for constructing the confidence set. Both the above methods achieve $\tilde{\mathcal{O}}(d\sqrt{H^3T})$ regret bound, where $H$ is the episode length , $d$ is the feature dimension, and $T$ is the total steps. Later, (Zhou et al., 2021a) provides a new tail inequality and adapts the weighted ridge regression to UCRL-VTR to improve the regret bound $\tilde{\mathcal{O}}(dH\sqrt{T})$. Moreover, (Yang & Wang, 2020) studies an interesting type of linear mixture MDPs that are bilinear in two feature embeddings and proposes MatrixRL to achieve $\tilde{\mathcal{O}}(dH^2\sqrt{T})$ regret bound. More recently, (He et al., 2022) proposes to improve OPPO with a new Bernstein-type bonus and achieve a near-optimal $\tilde{\mathcal{O}}(dH\sqrt{T})$ regret.

- *Linear MDPs*: Another related but different type of linear models is the linear MDP, where the transition model takes the form of the inner product between a state-action feature vector and the vector of unknown measures over states. For instance, (Jin et al., 2020) presents an optimistic variant of Least-Squares Value Iteration that achieves $\tilde{\mathcal{O}}(\sqrt{d^3H^3T})$ regret. (Wang et al., 2019) studies another related but more general class of MDPs with generalized linear function approximation and an optimistic closure assumption and presents value-based approaches with $\tilde{\mathcal{O}}(H\sqrt{d^3T})$ regret bound.

- *General model classes*: Recently, there are several works that address episodic MDPs under general function approximation, where a class of possible transition models is given to the algorithm. For instance, (Zhang, 2022) proposes a variant of Thompson sampling to favor models with high rewards for more aggressive exploration. Later, (Zhong et al., 2022) presents a new complexity measure, namely the generalized eluder coefficient, and proposes a variant of posterior sampling algorithm under a general model class.

**Infinite-Horizon Discounted MDPs With Function Approximation**. Without the restart capability of episodic MDPs, infinite-horizon discounted MDPs pose a unique challenge of tackling planning and exploration in the same single trajectory. As a result, the theoretical understanding of this setting under function approximation remains limited. For example, under linear MDPs, (Yang & Wang, 2019) proposes a variant of Q-learning that requires $\tilde{\mathcal{O}}(d/(1-\gamma)^2\epsilon^2)$ samples to find an $\epsilon$-optimal policy, where $\gamma$ is the discount factor. Under the linear mixture MDPs, (Zhou et al., 2021b) proposes the UCLK algorithm, which takes into consideration the confidence set of the least-square estimator of the model parameter and establishes a regret upper bound of $\tilde{\mathcal{O}}(d\sqrt{T}/(1-\gamma)^2)$. Subsequently, (Zhou et al., 2021a) introduces an improved version of UCLK, called UCLK$^+$, by incorporating the weighted ridge regression into the original UCLK and achieves a regret bound that matches the lower bound of $\tilde{\mathcal{O}}(d\sqrt{T}/(1-\gamma)^{1.5})$ established by (Zhou et al., 2021b). On the other hand, (Chen et al., 2022) also provides a variant of UCLK, namely UPAC-UCLK, which achieves $\tilde{\mathcal{O}}(d\sqrt{T}/(1-\gamma)^2) + \tilde{\mathcal{O}}(\sqrt{T}/(1-\gamma)^3)$ regret bound along with uniform-PAC sample complexity guarantee. However, the above UCLK-based approaches are computationally inefficient as they all require the costly extended value iteration for each state-action pair in each policy update, and this is already not tractable in MDPs of moderate sizes. Our work falls in this category and presents VBMLE as a computationally efficient solution to infinite-horizon linear mixture MDPs.

## 3    PROBLEM FORMULATION

**Markov Decision Processes (MDP) and Linear Feature Mapping.** An MDP is denoted by $\mathcal{M} := \langle \mathcal{S}, \mathcal{A}, P, R, T, \mu_0 \rangle$, where $\mathcal{S}$ and $\mathcal{A}$ represent the state and action spaces, respectively, $P$ is the dynamic model, $R : \mathcal{S} \times \mathcal{A} \to [0, 1]$ is the reward function, $T$ is the time horizon, and $\mu_0$ is the initial state distribution with $\mu_0(s) > 0$[1]. A linear mixture MDP is defined by the following:

---

[1]As there is a policy that achieves optimal value for *all* initial states $s$, or equivalently, for *all* initial distributions $\mu_0$, without loss of generality, it is common to take a strictly positive initial distribution.

- There exist an unknown parameter $\boldsymbol{\theta}^* \in \mathbb{R}^d$, and a known feature mapping $\phi(\cdot|\cdot, \cdot)$ : $\mathcal{S} \times \mathcal{A} \times \mathcal{S} \to \mathbb{R}^d$, such that $P(s'|s, a) = \langle \phi(s'|s, a), \boldsymbol{\theta}^* \rangle, \forall s', s \in \mathcal{S}, a \in \mathcal{A}$.

- $\|\boldsymbol{\theta}^*\|_2 \le \sqrt{d}$ and $\|\phi(s'|s, a)\|_2 \le L, \forall s', s \in \mathcal{S}, a \in \mathcal{A}$.

Moreover, let $\mathbb{P}$ denote the set of parameters that correspond to the product of the simplices for each (state, action) pair:

$$\mathbb{P} := \left\{ \boldsymbol{\theta} : 0 \le \langle \phi(\cdot|s, a), \boldsymbol{\theta} \rangle \le 1, \sum_{s' \in \mathcal{S}} \langle \phi(s'|s, a), \boldsymbol{\theta} \rangle = 1, \forall s \in \mathcal{S}, a \in \mathcal{A} \right\}, \tag{1}$$

where $\boldsymbol{\theta}$ denotes the parameter of the transition dynamics model and $\phi(\cdot|s, a)$ is the known feature mapping function.

A policy $\pi : \mathcal{S} \to \Delta(\mathcal{A})$, where $\Delta(\mathcal{A})$ is the set of all probability distributions on $\mathcal{A}$, designed to maximize the sum of discounted reward, which is denoted by the value function:

$$V^\pi(s; \boldsymbol{\theta}) := \mathbb{E}_{\substack{a_i \sim \pi(\cdot|s_i) \\ s_{i+1} \sim \langle \phi(\cdot|s_i, a_i), \boldsymbol{\theta} \rangle}} \left[ \sum_{i=0}^\infty \gamma^i r(s_i, a_i) \Big| s_0 = s \right]. \tag{2}$$

Similarly, the action value function $Q^\pi(s, a; \boldsymbol{\theta})$ is defined as

$$Q^\pi(s, a; \boldsymbol{\theta}) := \mathbb{E}_{\substack{a_i \sim \pi(\cdot|s_i) \\ s_{i+1} \sim \langle \phi(\cdot|s_i, a_i), \boldsymbol{\theta} \rangle}} \left[ \sum_{i=0}^\infty \gamma^i r(s_i, a_i) \Big| s_0 = s, a_0 = a \right]. \tag{3}$$

Moreover, we let $J(\pi; \boldsymbol{\theta}) := \mathbb{E}_{s \sim \mu_0}[V^*(s; \boldsymbol{\theta}^*)]$ denote the mean reward achievable for the MDP with parameter $\boldsymbol{\theta}$ under policy $\pi$ over the initial probability distribution $\mu_0$.

**Optimal Value and Regret.** We then define the optimal value function to be the maximum value obtained by a policy: $V^*(s; \boldsymbol{\theta}) = \max_\pi V^\pi(s; \boldsymbol{\theta})$. In the discounted linear mixture MDP setting (Zhou et al., 2021b), the cumulative regret $\mathcal{R}(T)$ for the MDP with parameter $\boldsymbol{\theta}^*$ is defined to be the total difference of value function between the optimal policy and the learned policy $\pi_t$, where

$$\mathcal{R}(T) := \sum_{t=1}^T \left[ V^*(s_t; \boldsymbol{\theta}^*) - V^{\pi_t}(s_t, \boldsymbol{\theta}^*) \right], \quad s_1 \sim \mu_0. \tag{4}$$

Based on the fundamental result that there exists a policy that achieves optimal value for *all* states, we use $\pi^*(\boldsymbol{\theta})$ to denote an optimal policy with respect to a given model parameter $\boldsymbol{\theta}$ as

$$\pi^*(\boldsymbol{\theta}) := \operatorname*{argmax}_\pi J(\pi; \boldsymbol{\theta}). \tag{5}$$

# 4 VBMLE FOR LINEAR MDPS

**Introduction to the VBMLE Principle.** We now introduce the idea behind the classic value biasing principle. Consider first the certainty equivalence principle (Kumar & Varaiya, 2015) employing the straightforward Maximum Likelihood Estimate (MLE) as

$$\widehat{\boldsymbol{\theta}}_t := \operatorname*{argmax}_{\boldsymbol{\theta} \in \mathbb{P}} \left\{ \prod_{i=1}^{t-1} p(s_{i+1}|, s_i, a_i; \boldsymbol{\theta}) \right\}, \tag{6}$$

That is, at each time step $t$, the learner employs the policy $\pi_t^{\text{MLE}}$ that is optimal for the current estimate $\widehat{\boldsymbol{\theta}}_t$. Under appropriate technical conditions, it has been shown in (Borkar & Varaiya, 1979) that $\widehat{\boldsymbol{\theta}}_t$ converges almost surely to a random $\widehat{\boldsymbol{\theta}}_\infty$, for which

$$p(s'|s, \pi_\infty^{\text{MLE}}(s); \widehat{\boldsymbol{\theta}}_\infty) = p(s'|s, \pi_\infty^{\text{MLE}}(s); \boldsymbol{\theta}^*), \forall s, s' \in \mathcal{S}, \tag{7}$$

where $\pi_\infty^{\text{MLE}}$ represents the optimal policy corresponding to $\widehat{\boldsymbol{\theta}}_\infty$. This convergence property is called the *closed-loop identification* property; it means that asymptotically the transition probabilities resulting from the application of the policy $\pi_\infty^{\text{MLE}}$ are correctly estimated. An important consequence is that $J(\pi^*(\widehat{\boldsymbol{\theta}}_\infty); \widehat{\boldsymbol{\theta}}_\infty) = J(\pi^*(\widehat{\boldsymbol{\theta}}_\infty); \boldsymbol{\theta}^*)$. Since $\pi_\infty^{\text{MLE}}$ is not necessarily optimal for $\boldsymbol{\theta}^*$, this implies

$$J(\pi^*(v); \widehat{\boldsymbol{\theta}}_\infty) \le J(\pi^*(\boldsymbol{\theta}^*); \boldsymbol{\theta}^*). \tag{8}$$

---

**Algorithm 1** VBMLE for Reinforcement Learning in Linear Mixture MDPs

---

1: **Input:** $\alpha(t)$
2: **for** $t = 1, 2, \cdots$ **do**
3:      $\boldsymbol{\theta}_{\mathrm{t}}^{\mathbf{V}} := \mathrm{argmax}_{\boldsymbol{\theta} \in \mathbb{P}} \left\{ \sum_{i=1}^{t-1} \log \langle \phi(s_{i+1} | s_i, a_i), \boldsymbol{\theta} \rangle + \frac{\lambda}{2} \|\boldsymbol{\theta}\|_2^2 + \alpha(t) \cdot V^*(s_t; \boldsymbol{\theta}) \right\}.$
4:      $a_t = \underset{a \in \mathcal{A}}{\mathrm{argmax}} \, Q^*(s_t, a; \boldsymbol{\theta}_t^{\mathbf{R}})$
5: **end for**

---

The idea of the Value-Biased method is to try to undo the bias in (8) by adding a bias term that favors parameters with larger optimal total return. This leads to the principle of Value-Biased Maximum Likelihood Estimate (VBMLE) originally proposed in the adaptive control literature by (Kumar & Lin, 1982) as follows:

$$\boldsymbol{\theta}_t^{\mathbf{VBMLE}} := \underset{\boldsymbol{\theta} \in \mathbb{P}}{\mathrm{argmax}} \left\{ \sum_{i=1}^{t-1} \log p(s_{i+1} | s_i, a_i; \boldsymbol{\theta}) + \alpha(t) \cdot J(\pi^*(\boldsymbol{\theta}); \boldsymbol{\theta}) \right\}, \tag{9}$$

where $\alpha(t)$ is a positive increasing sequence that weights the bias in favor of parameters with larger total return. VBMLE employs this biasing method to handle the exploration-exploitation trade-off.

**VBMLE for Discounted Linear Mixture MDPs.** In this paper, we adapt the VBMLE principle to the RL problem in the linear mixture MDP setting. Specifically, at each step, the learner would (i) choose the parameter estimate that maximizes the regularized log-likelihood plus the value-bias as

$$\boldsymbol{\theta}_{\mathrm{t}}^{\mathbf{V}} := \underset{\boldsymbol{\theta} \in \mathbb{P}}{\mathrm{argmax}} \left\{ \sum_{i=1}^{t-1} \log \langle \phi(s_{i+1} | s_i, a_i), \boldsymbol{\theta} \rangle + \frac{\lambda}{2} \|\boldsymbol{\theta}\|_2^2 + \alpha(t) \cdot V^*(s_t; \boldsymbol{\theta}) \right\}, \tag{10}$$

where $\lambda$ is a positive constant for regularization, and then (ii) employ an optimal policy with respect to $\boldsymbol{\theta}_{\mathrm{t}}^{\mathbf{V}}$. Notice that the term $V^*(s_t; \boldsymbol{\theta})$ can be computed by using the standard Value Iteration presented as Algorithm 2 in Appendix. If there are multiple maximizers for (10), then one could break the tie arbitrarily. For clarity, we also summarize the procedure of VBMLE in Algorithm 1.

**Features of VBMLE for Linear Mixture MDPs.** We highligh the salient features of the VBMLE method in Algorithm 1 as follows.

- **Computational Efficiency:** As mentioned earlier, UCLK (Zhou et al., 2021b) suffers from high computational complexity as it requires computing an estimate of the model parameter for *each* state-action pair in each iteration. This renders UCLK intractable when either the state space or the action space is large. By contrast, the VBMLE approach, which applies value-bias to guide the exploration under the MLE, only requires solving one single maximization problem for the dynamics model parameter $\boldsymbol{\theta}$ in each iteration, making it computationally efficient and superior. Accordingly, VBMLE could serve as a more computationally feasible algorithm for RL in linear mixture MDPs in practice.

- **VBMLE is Parameter-Free:** As shown in Algorithm 1, the only parameter required by VBMLE is $\alpha(t)$, which determines the weight of the value bias. As will be shown in Section 5, one could simply choose $\alpha(t) = \sqrt{t}$ to achieve the required regret bound, and moreover this simple choice also leads to superior empirical regret performance. As a result, VBMLE is parameter-free and therefore does not require any hyperparameter tuning.

- **Distributional Perspective:** In contrast to the existing RL methods for linear mixture MDPs (Ayoub et al., 2020; Zhou et al., 2021a;b; Chen et al., 2022) that aim to learn the unknown parameter via regression on the value function (or termed *value-targeted regression*), the proposed VBMLE takes a distributional perspective through directly learning the whole collection of transition probabilities through value-biased maximum likelihood estimation. This perspective has also been adopted by the prior works on applying VBMLE to the contextual bandit problems (Hung et al., 2021; Hung & Hsieh, 2023).

We also highlight the differences between VBMLE for RL and VBMLE for bandits in Appendix C.

**Remark 1.** Due to the non-concavity of VBMLE, we propose to solve VBMLE by Bayesian optimization (BO), which is a powerful and generic method for provably maximizing (possibly non-concave) black-box objective functions. As a result, BO can provably find an $\epsilon$-optimal solution to VBMLE within finite iterations. Specifically:

- We have applied the GP-UCB algorithm, which is one classic BO algorithm and has been shown to provably find an $\epsilon$-optimal solution within $\tilde{\mathcal{O}}(1/\epsilon^2)$ iterations under smooth (possibly non-concave) objective functions (Srinivas et al., 2012). Each sample taken by GP-UCB requires only one run of standard Value Iteration in Algorithm 2.

- To further demonstrate the compatibility of VBMLE and BO, we have extended the regret analysis of VBMLE to the case where only an $\epsilon$-optimal VBMLE solution is obtained. Specifically, let $H$ denote the number of samples taken by GPUCB in each maximization run of finding VBMLE. We show that VBMLE augmented with BO can achieve sub-linear regret as shown in Theorem 4. By using a moderate $H$, one could easily recover the same regret bound as that of VBMLE with an exact maximizer. In our experiments, we find that choosing $H = 25$ is sufficient and also computationally efficient.

- The complexity of VBMLE with GP-UCB for finding $\boldsymbol{\theta}_t^{\mathbf{V}}$ is to solve the standard Value Iteration for only $H + 1$ times ($H$ is for BO, each sample requires one value iteration in our objective function, and another 1 for value iteration for $\boldsymbol{\theta}_t^{\mathbf{V}}$). This is a clear computational advantage over the EVI in UCLK.

## 5 REGRET ANALYSIS

In this section, we formally present the regret analysis of the VBMLE algorithm. To begin with, we introduce the following useful notations:

$$\ell_t(\boldsymbol{\theta}) := \sum_{i=1}^{t-1} \log\langle\phi(s_{i+1}|s_i, a_i), \boldsymbol{\theta}\rangle + \frac{\lambda}{2}\|\boldsymbol{\theta}\|_2^2 \tag{11}$$

$$\boldsymbol{\theta}_t^{\mathrm{MLE}} := \underset{\boldsymbol{\theta}\in\mathbb{P}}{\operatorname{argmax}}\, \ell_t(\boldsymbol{\theta}), \tag{12}$$

$$\mathbf{A}_t := \sum_{i=1}^{t-1} \phi(s_{i+1}|s_i, a_i)\phi(s_{i+1}|s_i, a_i)^\top + \lambda I. \tag{13}$$

If there are multiple maximizers for (12), then one could break the tie arbitrarily.

**Assumption 1.** *The following information for the transition probability $P$ is known:*

- *The set of zero transition $P_0 := \{(s, a, s')|P(s'|s, a) = 0, \forall s, s' \in \mathcal{S}, a \in \mathcal{A}\}$.*

- *The lower bound non-zero transition probabilities $p_{min} := \min_{(s,a,s')\notin P_0} P(s'|s, a)$.*

*We then redefine the probability simplex based on the above assumption as follows:*

$$\mathbb{P} := \left\{ \boldsymbol{\theta} \,\middle|\, p_{min} \le \langle\phi(s'|s, a), \boldsymbol{\theta}\rangle \le 1, \forall(s, a, s') \notin P_0; \right.$$
$$\langle\phi(s'|s, a), \boldsymbol{\theta}\rangle = 0, \forall(s, a, s') \in P_0;$$
$$\left. \sum_{s'\in\mathcal{S}} \langle\phi(s'|s, a), \boldsymbol{\theta}\rangle = 1, \forall s \in \mathcal{S}, a \in \mathcal{A} \right\}. \tag{14}$$

**Remark 2.** This assumption suggests that the magnitude of the gradient of the log probability for the observed transition, denoted as $\|\nabla_{\boldsymbol{\theta}} \log\langle\phi(s_{i+1}|s_i, a_i), \boldsymbol{\theta}_t^{\mathrm{MLE}}\rangle\|_2$, is bounded from above. A similar assumption is made in (Kumar & Lin, 1982; Mete et al., 2021). In some scenarios, the knowledge of $p_{\min}$ may not be readily available. To address this, we introduce an alternative version of VBMLE, termed *Adaptive VBMLE*, to address this issue. This variant employs the following

adaptive constraint, resembling a probability simplex, to solve $\boldsymbol{\theta}_t^{\mathbf{V}}$.

$$
\mathbb{P}_t := \left\{ \boldsymbol{\theta} \,\middle|\, \frac{1}{\log t} \le \langle \phi(s'|s,a), \boldsymbol{\theta} \rangle \le 1, \forall (s,a,s') \notin P_0; \right.
$$
$$
\langle \phi(s'|s,a), \boldsymbol{\theta} \rangle = 0, \forall (s,a,s') \in P_0;
$$
$$
\left. \sum_{s' \in \mathcal{S}} \langle \phi(s'|s,a), \boldsymbol{\theta} \rangle = 1, \forall s \in \mathcal{S}, a \in \mathcal{A} \right\}. \tag{15}
$$

The regret bound for this variant is detailed in Theorem 3.

## 5.1 Convergence Analysis of MLE in Linear Mixture MDPs

To begin with, we highlight the main technical challenges as follows: A natural idea is to leverage the Azuma–Hoeffding inequality on the log-likelihood ratio: $\ell_t(\boldsymbol{\theta}_t^{\mathrm{MLE}}) - \ell_t(\boldsymbol{\theta}^*)$, and then find the distance between $\boldsymbol{\theta}_t^{\mathrm{MLE}}$ and the true parameter $\boldsymbol{\theta}^*$. However, it is known that the stochastic process induced by the maximum log-likelihood ratio is actually a *sub-martingale* (shown in Lemma 6 in Appendix for completeness). To address these issue, we propose several novel techniques: (i) We first propose to construct a novel super-martingale (cf. Lemma 1) to characterize the convergence rate of the MLE in linear mixture MDPs, which could be of independent interest beyond RL problems. Interestingly, this supermartingale consists of a term that could be interpreted as the regret in the online portfolio selection problem and thereby offers an interesting connection between linear mixture MDPs and online learning. (ii) Built on (i), to utilize Azuma-Hoeffding inequality, we need to carefully handle the sum of squared supermartingale differences, which do not have an explicit uniform upper bound and require a more sophisticated argument.

To begin with, we provide several useful definitions as follows. Define the likelihood ratio as

$$
L_t(\boldsymbol{\theta}) := \prod_{i=1}^{t-1} \frac{\Pr(s_{i+1}|s_i, a_i; \boldsymbol{\theta})}{\Pr(s_{i+1}|s_i, a_i; \boldsymbol{\theta}^*)} \cdot \exp\left( \frac{\lambda}{2} \|\boldsymbol{\theta}\|_2^2 \right). \tag{16}
$$

We proceed to construct two useful helper stochastic processes as follows: For each $t \in \mathbb{N}$,

$$
X_t := \ell_t(\boldsymbol{\theta}_{t-1}^{\mathrm{MLE}}) - \ell_t(\boldsymbol{\theta}^*) + \sum_{i=1}^{t-1} z_i, \tag{17}
$$

$$
z_t := \ell_t(\boldsymbol{\theta}_t^{\mathrm{MLE}}) - \ell_t(\boldsymbol{\theta}_{t-1}^{\mathrm{MLE}}). \tag{18}
$$

**Lemma 1.** *For all $\lambda \ge 0$, the stochastic process $\{ L_t(\boldsymbol{\theta}_{t-1}^{\mathrm{MLE}}) \cdot \prod_{i=1}^{t-1} \exp(-z_i) \}$ is a martingale, i.e.,*

$$
\mathbb{E}_{s_{t+1} \sim \Pr(\cdot|s_t, a_t; \boldsymbol{\theta}^*)} \left[ L_{t+1}(\boldsymbol{\theta}_t^{\mathrm{MLE}}) \cdot \prod_{i=1}^{t} \exp(-z_i) \,\middle|\, \mathcal{F}_t \right] = L_t(\boldsymbol{\theta}_{t-1}^{\mathrm{MLE}}) \cdot \prod_{i=1}^{t-1} \exp(-z_i), \tag{19}
$$

*where $\mathcal{F}_t := \{ s_1, a_1, \cdots, s_t, a_t \}$ denotes the causal information up to time $t$.*

**Corollary 1.** *For all $\lambda \ge 0$, the stochastic process $\{X_t\}$ is a supermartingale, i.e.,*

$$
\mathbb{E}_{s_{t+1} \sim \Pr(\cdot|s_t, a_t; \boldsymbol{\theta}^*)} \left[ \ell_{t+1}(\boldsymbol{\theta}_t^{\mathrm{MLE}}) - \ell_{t+1}(\boldsymbol{\theta}^*) - \sum_{i=1}^{t} z_i \,\middle|\, \mathcal{F}_t \right] \le \ell_t(\boldsymbol{\theta}_{t-1}^{\mathrm{MLE}}) - \ell_t(\boldsymbol{\theta}^*) - \sum_{i=1}^{t-1} z_i. \tag{20}
$$

*This corollary can be proved by applying Jensen's inequality to (19),*

Notably, Corollary 1 offers a useful insight that a supermartingale that involves the log-likelihood ratio could still be constructed despite that $\ell_t(\boldsymbol{\theta}_t^{\mathrm{MLE}}) - \ell_t(\boldsymbol{\theta}^*)$ is a *submartingale*. This result generalizes the classic result in (Kumar & Lin, 1982, Lemma 3) for tabular MDPs to the linear mixture MDP setting, and is also holds for non-regularized MLE ($\lambda = 0$). To establish Theorem 1, we define a useful quantity $\Delta_t$ as

$$
\Delta_t := \sum_{i=1}^{t-1} z_i = \sum_{i=1}^{t-1} \log\left( \frac{\phi_i(s_{i+1})^\top \boldsymbol{\theta}_t^{\mathrm{MLE}}}{\phi_i(s_{i+1})^\top \boldsymbol{\theta}_i^{\mathrm{MLE}}} \right), \tag{21}
$$

where $\phi_i(s) := \phi(s|s_i, a_i)$ is a shorthand for the feature vector. In the following lemma, we present an upper bound for $\Delta_t$. Recall that $\|\phi(s'|s,a)\|_2 \le L$, for all $s, s' \in \mathcal{S}$ and $a \in \mathcal{A}$.

**Lemma 2.** *For all $\lambda \geq 0$, we have*

$$\Delta_t \leq \frac{8d^2}{p_{min}^2} \log \left( \frac{d\lambda + (t-1)L^2}{d} \right). \tag{22}$$

**Remark 3** (Connection between linear mixture MDPs and online learning). Through $\Delta_t$ and Lemma 2, we could build an interesting connection between MLE in linear mixture MDPs and the *Follow-the-Leader* algorithm (Gaivoronski & Stella, 2000) in online learning. The connection is two-fold: (i) *MLE in linear mixture MDPs can be viewed as a variant of online portfolio selection problem*: We find that the MLE optimization problem in linear mixture MDPs takes the same form as the classic online portfolio selection problem (Hazan et al., 2016). Specifically, the feature vectors and the dynamics model parameter in linear mixture MDPs correspond to the price vectors and the asset allocation, respectively. The main difference of the two problems lies in the feasible set and the constraints. (ii) *Iterative MLE is equivalent to Follow-the-Leader algorithm*: Another interesting connection is that applying MLE in each time step would correspond to the classic Follow-the-Leader algorithm (Gaivoronski & Stella, 2000). Moreover, the term $\Delta_t$ in 22 could be interpreted as the regret of the Follow-the-Leader algorithm in online learning. With that said, one could verify that Lemma 2 is consistent with the regret quantified in (Gaivoronski & Stella, 2000).

Based on the supporting lemmas introduced above, we are ready to formally present the convergence result of MLE in linear mixture MDPs.

**Theorem 1.** *With probability at least $1 - \delta$, we have*

$$\|\boldsymbol{\theta}^* - \boldsymbol{\theta}_t^{\text{MLE}}\|_{\mathbf{A}_t}^2 \leq \frac{37d^2}{p_{min}^2} \cdot \log \left( \frac{d\lambda + tL^2}{d} \right) \cdot \log \frac{1}{\delta}, \tag{23}$$

The complete proof of Theorem 1 is provided in Appendix B.1, and here we provide a proof sketch:

*Proof Sketch.* Based on the result in Lemma 1, we can apply Azuma–Hoeffding inequality presented in Lemma 5 to get the high probability bound of log-likelihood ratio. There are two main challenges that need to be handled: (i) The first one is the additional term $\Delta_t$, we find a connection to the analysis of the online portfolio selection problem and use a similar approach to handle it. (ii) The other one is $M_t$, which represents the cumulative difference of the super-martingale. We adopt a similar approach by considering a stopping time to ensure that this theorem holds with high probability.

## 5.2 REGRET BOUND OF VBMLE

In this subsection, we formally provide the regret bound of the proposed VBMLE algorithm.

**Theorem 2.** *For all linear mixture MDP $\mathcal{M} = \langle \mathcal{S}, \mathcal{A}, P, R, T, \mu_0 \rangle$, with probability at least $1 - \frac{1}{T} - 3\delta$ and choosing $\alpha(t) = \sqrt{t}$, VBMLE in Algorithm 1 has a regret upper bound as*

$$\mathcal{R}(T) = \mathcal{O} \left( \frac{d\sqrt{T} \log T}{p_{min}^4 (1-\gamma)^2} \right). \tag{24}$$

The complete proof of Theorem 2 is provided in Appendix B.2, and here we provide a proof sketch:

1. Similar to the analysis of the upper-confidence bound approach, which uses the concentration inequality to replace the term associated with an optimal policy, under VBMLE we can replace $V^*(s_t, \boldsymbol{\theta}^*)$ by applying the objective function of VBMLE.

2. Then, there are two terms that need to be handled: (i) $\|\boldsymbol{\theta}_t^{\text{MLE}} - \boldsymbol{\theta}^*\|_{\mathbf{A}_t}$ and (ii) $\|\boldsymbol{\theta}_t^{\mathbf{V}} - \boldsymbol{\theta}^*\|_{\mathbf{A}_t}$. We provide a novel theorem of the confidence ellipsoid of the maximum likelihood estimator in linear mixture MDPs in Theorem 1 to deal with (i).

3. In contrast to the regret analysis presented in (Hung et al., 2021), where the likelihood of an exponential family distribution was considered, analyzing regret in the linear mixture MDP setting is more complex due to the absence of simple closed-form expressions for both $\boldsymbol{\theta}_t^{\text{MLE}}$ and $\boldsymbol{\theta}_t^{\mathbf{V}}$. Additionally, in this context, the bias term is not linear with respect to $\boldsymbol{\theta}$, even if we represent it as $Q^*(s_t, a_t; \boldsymbol{\theta}) = \langle \sum_{s' \in \mathcal{S}} \phi(s'|s_t, a_t) V^*(s_t, \boldsymbol{\theta}), \boldsymbol{\theta} \rangle$. To address these challenges, we adopt a novel approach by completing the square of $\|\boldsymbol{\theta}_t^{\mathbf{V}} - \boldsymbol{\theta}^*\|_{\mathbf{A}_t}$ and successfully overcome the problems mentioned above.

We also provide the regret analysis for this variant of VBMLE in Appendix D.

## 6 NUMERICAL EXPERIMENTS

We demonstrate the empirical performance of VBMLE in terms of both regret and computation time in this section. we conduct experiments on a simple environment with discrete state and action spaces. To provide a detailed understanding of how we transition from the tabular MDPs to the linear mixture MDP setting, we have outlined the procedure in the Appendix F.1. The following result includes a comparison between VBMLE, UCLK (Zhou et al., 2021b) and UCLK$^+$ (Zhou et al., 2021a), which are both well-known algorithms used in the context of infinite-horizon linear mixture MDPs. Another baseline algorithm is PSRL (Osband et al., 2013), a popular benchmark method for tabular RL. Details regarding the selected hyperparameters can be found in Appendix F.2. In the following

- **Empirical Regret:** Figure 1 provides the empirical regret of VBMLE, UCLK, UCLK$^+$ and PSRL across various sizes of linear MDPs, and the results demonstrate that the two varients, where VBMLE (TR) is VBMLE with trust-region constrained algorithm for solving $\boldsymbol{\theta}_t^{\mathbf{V}}$ and VBMLE (BO) is VBMLE with GP-UCB for optimization, outperform the other baselines in terms of regret performance. In Figure 1(b), only the results of VBMLE with GP-UCB proposed in Appendix E and PSRL are presented, as the other approaches are intractable for the larger MDP ($|\mathcal{S}| = 100$) of the MDP. The result shows that though PSRL has sub-linear regret in small-scale MDP ($|\mathcal{S}| = 5$), it has linear regret due to that it does not leverage the structure of linear feature mapping. We also provide the standard deviation of the regret at the final step in Table G.1. VBMLE also has better robustness with an order of magnitude smaller standard deviation than UCLK.

- **Computation Time:** UCLK requires $U|\mathcal{S}||\mathcal{A}|$ times of solving the constrained quadratic optimization problem per step, where $U$ is the times of value iteration, and VBMLE with BO only requires once $K + 1$ times, where $K$ is the time horizon for BO. Table 6 displays the computation time per step within the same environment as depicted in Figure 1(a). It is evident that the computational complexity of UCLK and UCLK$^+$ render the algorithm impractical in large MDPs.

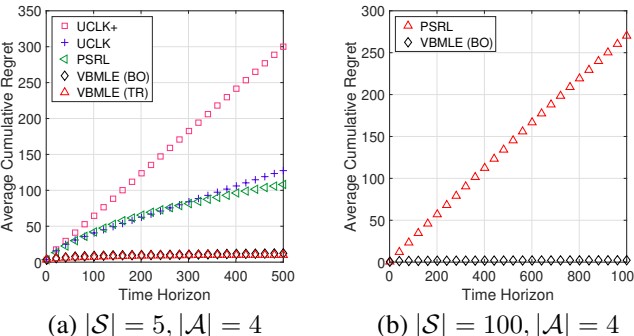

(a) $|\mathcal{S}| = 5, |\mathcal{A}| = 4$        (b) $|\mathcal{S}| = 100, |\mathcal{A}| = 4$

Figure 1: Regret averaged over 10 trials.

Table 1: Computation time per step under different sizes of linear mixture MDPs.

|  | $|\mathcal{S}| = 3, |\mathcal{A}| = 2$ | $|\mathcal{S}| = 5, |\mathcal{A}| = 4$ | $|\mathcal{S}| = 15, |\mathcal{A}| = 4$ | $|\mathcal{S}| = 100, |\mathcal{A}| = 4$ |
|---|---|---|---|---|
| **VBMLE (TR)** | 0.793s | 2.359s | 42.232s | - |
| **VBMLE (BO)** | 2.06s | 2.232s | 3.999s | 19.687s |
| **UCLK** | 3.135s | 49.763s | $\geq 35$hr | - |
| **UCLK$^+$** | 0.741s | 20.128s | $\geq 37$hr | - |

## 7 CONCLUSION

We proposed a provably effective and computationally efficient algorithm for solving linear MDPs, called VBMLE. The sample complexity of the proposed is proved to be upper bounded by $\mathcal{O}\left(d\sqrt{T}\log T/(p_{\min}^4(1-\gamma)^2)\right)$. The proposed algorithm is different from the existing value-target regression approach and leverages the MLE with value bias to learn the dynamic. We provide a novel theorem to show the confidence ellipsoid of MLE and the simulation result demonstrates the empirical performance of VBMLE.

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

# A    PROOFS OF THE SUPPORTING LEMMAS

Recall that

$$L_t(\boldsymbol{\theta}) = \prod_{i=1}^{t-1} \frac{\Pr(s_{i+1}|s_i, a_i; \boldsymbol{\theta})}{\Pr(s_{i+1}|s_i, a_i; \boldsymbol{\theta}^*)} \cdot \exp\left(\frac{\lambda}{2}\|\boldsymbol{\theta}\|_2^2\right) \tag{25}$$

$$X_t = \ell_t(\boldsymbol{\theta}_{t-1}^{\mathrm{MLE}}) - \ell_t(\boldsymbol{\theta}^*) + \sum_{i=1}^{t-1} z_i, \tag{26}$$

$$z_t = \ell_t(\boldsymbol{\theta}_t^{\mathrm{MLE}}) - \ell_t(\boldsymbol{\theta}_{t-1}^{\mathrm{MLE}}), \tag{27}$$

we then introduce some supporting lemmas for the proof of regret.

**Lemma 3** (Lemma C.2 in (Zhou et al., 2021b), Lemma 11 in (Abbasi-Yadkori et al., 2011)). *Given any $\{\phi(s_{t+1}|s_t, a_t)\}_{t=1}^T \in \mathbb{R}^d$ satisfying that $\|\phi(s_{t+1}|s_t, a_t)\|_2 \le L$. For all $\lambda > 0$, we have*

$$\sum_{t=1}^T \|\phi(s_{t+1}|s_t, a_t)\|_{\mathbf{A}_t^{-1}}^2 \le 2d\log\left(\frac{d\lambda + TL^2}{d}\right). \tag{28}$$

**Lemma 4.** *$\forall s \in \mathcal{S}, a \in \mathcal{A}$, and $\boldsymbol{\theta} \in \mathbb{P}$, we have $Q^*(s, a, \boldsymbol{\theta}) \le \frac{1}{1-\gamma}$.*

Lemma 4 is a direct result of that the reward function is bounded by 1, i.e., $R(\cdot, \cdot) \le 1$.

**Lemma 5** (Azuma–Hoeffding Inequality). *Suppose $\{X_1, X_2, \cdots\}$ is a martingale or super-martingale. Then for all positive integers $t$ and all positive reals $\delta$, we have*

$$\Pr\left(X_t - X_0 \ge \sqrt{2M_t \log\frac{1}{\delta}}\right) \le \delta, \tag{29}$$

*where $M_t := \sum_{i=1}^t (X_t - X_{t-1})^2$.*

**Lemma 6.**

$$L_t(\boldsymbol{\theta}_t^{\mathrm{MLE}}) := \max_{\boldsymbol{\theta} \in \mathbb{P}} L_t(\boldsymbol{\theta}) \tag{30}$$

*is a sub-martingale.*

*Proof.* Given the causal information $\mathcal{F}_t$, we have

$$\mathbb{E}_{s_{t+1}\sim\Pr(\cdot|s_t,a_t;\boldsymbol{\theta}^*)}\left[\max_{\boldsymbol{\theta}\in\mathbb{P}} L_{t+1}(\boldsymbol{\theta})\big|\mathcal{F}_t\right] \ge \max_{\boldsymbol{\theta}\in\mathbb{P}}\mathbb{E}_{s_{t+1}\sim\Pr(\cdot|s_t,a_t;\boldsymbol{\theta}^*)}\left[L_{t+1}(\boldsymbol{\theta})\big|\mathcal{F}_t\right] \tag{31}$$

$$= \max_{\boldsymbol{\theta}\in\mathbb{P}}\left\{\sum_{s'\in\mathcal{S}}\Pr(s'|s_t,a_t;\boldsymbol{\theta}^*)\cdot\frac{\Pr(s'|s_t,a_t;\boldsymbol{\theta})}{\Pr(s'|s_t,a_t;\boldsymbol{\theta}^*)}\cdot L_t(\boldsymbol{\theta})\right\} \tag{32}$$

$$= L_t(\boldsymbol{\theta}_t^{\mathrm{MLE}}), \tag{33}$$

where (31) holds by $\max_{\boldsymbol{\theta}\in\mathbb{P}} L_{t+1}(\boldsymbol{\theta}) \ge L_{t+1}(\boldsymbol{\theta}')$, and

$$\boldsymbol{\theta}' := \operatorname*{argmax}_{\boldsymbol{\theta}\in\mathbb{P}}\left\{\mathbb{E}_{s_{t+1}\sim\Pr(\cdot|s_t,a_t;\boldsymbol{\theta}^*)}\left[L_{t+1}(\boldsymbol{\theta})\big|\mathcal{F}_t\right]\right\}. \tag{34}$$

$\square$

**Remark 4.** The lemma shows that it is not able to apply Azuma–Hoeffding inequality on the maximum likelihood ratio. However, we can still add an additional term to construct a supermartingale:

$$\mathbb{E}_{s_{t+1}\sim\Pr(\cdot|s_t,a_t;\boldsymbol{\theta}^*)}\left[\max_{\boldsymbol{\theta}\in\mathbb{P}} L_{t+1}(\boldsymbol{\theta})\cdot\prod_{i=1}^t\left(\sum_{s'\in\mathcal{S}} h_t(s')\right)^{-1}\big|\mathcal{F}_t\right] = L_t(\boldsymbol{\theta}_t^{\mathrm{MLE}})\cdot\prod_{i=1}^{t-1}\left(\sum_{s'\in\mathcal{S}} h_t(s')\right)^{-1} \tag{35}$$

where

$$h_t(s') := \max_{\boldsymbol{\theta} \in \mathbb{P}} \left\{ \frac{L_t(\boldsymbol{\theta})}{L_t(\boldsymbol{\theta}_t^{\text{MLE}})} \cdot \Pr(s'|s_t, a_t; \boldsymbol{\theta}) \right\}. \tag{36}$$

It's important to note that there are two primary challenges when dealing with $h_t(s')$:

- Dependency on state space size: The first challenge arises from the fact that $\sum_{s' \in \mathcal{S}} h_t(s')$ exhibits a dependency on the size of the state space. This dependence can complicate the regret analysis, especially in cases where the state space is large.

- State-dependent maximization: The second challenge is related to the maximization inside $h_t(s')$, which is also influenced by the specific state $s'$ under consideration. Formally, the absence of a closed-form expression for this maximization makes it challenging to conduct a straightforward analysis.

To address the above two challenges, we then introduce a new supermartingale associate to the likelihood ratio evaluated on the previous step's maximum likelihood estimator.

## A.1 PROOF OF LEMMA 1

Recall that

$$X_t := \ell_t(\boldsymbol{\theta}_{t-1}^{\text{MLE}}) - \ell_t(\boldsymbol{\theta}^*) + \sum_{i=1}^{t-1} z_i, \tag{37}$$

$$z_t := \ell_t(\boldsymbol{\theta}_t^{\text{MLE}}) - \ell_t(\boldsymbol{\theta}_{t-1}^{\text{MLE}}). \tag{38}$$

For ease of exposition, we restate Lemma 1 as follows.

**Lemma.** *The stochastic process $\{L_t(\boldsymbol{\theta}_{t-1}^{\text{MLE}}) \cdot \prod_{i=1}^{t-1} \exp(-z_i)\}$ is a martingale, i.e.,*

$$\mathbb{E}_{s_{t+1} \sim \Pr(\cdot|s_t, a_t; \boldsymbol{\theta}^*)} \left[ L_{t+1}(\boldsymbol{\theta}_t^{\text{MLE}}) \cdot \prod_{i=1}^{t} \exp(-z_i) \Big| \mathcal{F}_t \right] = L_t(\boldsymbol{\theta}_{t-1}^{\text{MLE}}) \cdot \prod_{i=1}^{t-1} \exp(-z_i), \tag{39}$$

*where $\mathcal{F}_t := \{s_1, a_1, \cdots, s_t, a_t\}$ denotes the causal information up to time $t$.*

*Proof.* Recall the definition of likelihood ratio in (16). We have

$$\mathbb{E}_{s_{t+1} \sim \Pr(\cdot|s_t, a_t; \boldsymbol{\theta}^*)} \left[ L_{t+1}(\boldsymbol{\theta}_t^{\text{MLE}}) \cdot \prod_{i=1}^{t} \exp(-z_i) \Big| \mathcal{F}_t \right]$$

$$= \sum_{s' \in \mathcal{S}} \Pr(s'|s_t, a_t; \boldsymbol{\theta}^*) \cdot L_t(\boldsymbol{\theta}_t^{\text{MLE}}) \cdot \frac{\Pr(s'|s_t, a_t; \boldsymbol{\theta}_t^{\text{MLE}})}{\Pr(s'|s_t, a_t; \boldsymbol{\theta}^*)} \cdot \prod_{i=1}^{t} \exp(-z_i), \tag{40}$$

$$= L_t(\boldsymbol{\theta}_t^{\text{MLE}}) \cdot \prod_{i=1}^{t} \exp(-z_i), \tag{41}$$

$$= L_t(\boldsymbol{\theta}_{t-1}^{\text{MLE}}) \cdot \prod_{i=1}^{t-1} \exp(-z_i), \tag{42}$$

where (40) holds by the definition of the expectation, (41) holds due to $\boldsymbol{\theta}_t^{\text{MLE}} \in \mathbb{P}$ and $\mathcal{F}_t$-measurability, and (42) holds by $\exp(z_t) = \frac{L_t(\boldsymbol{\theta}_t^{\text{MLE}})}{L_t(\boldsymbol{\theta}_{t-1}^{\text{MLE}})}$. We complete this proof. $\qquad\square$

## A.2 PROOF OF LEMMA 2

Recall the definition of $\Delta_t$ from (21) as

$$\Delta_t := \sum_{i=1}^{t-1} \log \left( \frac{\phi_i(s_{i+1})^\top \boldsymbol{\theta}_t^{\text{MLE}}}{\phi_i(s_{i+1})^\top \boldsymbol{\theta}_i^{\text{MLE}}} \right) + \frac{\lambda}{2} \left( \|\boldsymbol{\theta}_t^{\text{MLE}}\|_2^2 - \|\boldsymbol{\theta}_0^{\text{MLE}}\|_2^2 \right), \tag{43}$$

where $\phi_i(s) := \phi(s|s_i, a_i)$ is a shorthand for the feature vector. For ease of exposition, we restate Lemma 2 as follows.

**Lemma.** *For all $\lambda \geq 0$, we have that*

$$\Delta_t \leq \frac{8d^2}{p_{min}^2} \log \left( \frac{d\lambda + (t-1)L^2}{d} \right). \tag{44}$$

*Proof.* To begin with, we have

$$\Delta_{t+1} = \sum_{i=1}^{t-1} \log \phi_i(s_{i+1})^\top \boldsymbol{\theta}_{t+1}^{\text{MLE}} - \sum_{i=1}^{t-1} \log \phi_i(s_{i+1})^\top \boldsymbol{\theta}_i^{\text{MLE}}$$
$$+ \frac{\lambda}{2} \left( \|\boldsymbol{\theta}_{t+1}^{\text{MLE}}\|_2^2 - \|\boldsymbol{\theta}_0^{\text{MLE}}\|_2^2 \right) + \log \frac{\phi_t(s_{t+1})^\top \boldsymbol{\theta}_{t+1}^{\text{MLE}}}{\phi_t(s_{t+1})^\top \boldsymbol{\theta}_t^{\text{MLE}}} \tag{45}$$

$$\leq \sum_{i=1}^{t-1} \log \phi_i(s_{i+1})^\top \boldsymbol{\theta}_t^{\text{MLE}} - \sum_{i=1}^{t-1} \log \phi_i(s_{i+1})^\top \boldsymbol{\theta}_i^{\text{MLE}}$$
$$+ \frac{\lambda}{2} \left( \|\boldsymbol{\theta}_t^{\text{MLE}}\|_2^2 - \|\boldsymbol{\theta}_0^{\text{MLE}}\|_2^2 \right) + \log \frac{\phi_t(s_{t+1})^\top \boldsymbol{\theta}_{t+1}^{\text{MLE}}}{\phi_t(s_{t+1})^\top \boldsymbol{\theta}_t^{\text{MLE}}} \tag{46}$$

$$\leq \sum_{i=1}^{t-1} \log \phi_i(s_{i+1})^\top \boldsymbol{\theta}_t^{\text{MLE}} - \sum_{i=1}^{t-1} \log \phi_i(s_{i+1})^\top \boldsymbol{\theta}_i^{\text{MLE}}$$
$$+ \frac{\lambda}{2} \left( \|\boldsymbol{\theta}_t^{\text{MLE}}\|_2^2 - \|\boldsymbol{\theta}_0^{\text{MLE}}\|_2^2 \right) + \frac{\langle \phi_t(s_{t+1}), \boldsymbol{\theta}_{t+1}^{\text{MLE}} - \boldsymbol{\theta}_t^{\text{MLE}} \rangle}{\phi_t(s_{t+1})^\top \boldsymbol{\theta}_t^{\text{MLE}}} \tag{47}$$

$$\leq \Delta_t + \left\| \frac{\phi_t(s_{t+1})}{\phi_t(s_{t+1})^\top \boldsymbol{\theta}_t^{\text{MLE}}} \right\|_{\mathbf{A}^{-1}} \cdot \|\boldsymbol{\theta}_{t+1}^{\text{MLE}} - \boldsymbol{\theta}_t^{\text{MLE}}\|_{\mathbf{A}}, \tag{48}$$

where (46) holds by $\boldsymbol{\theta}_t^{\text{MLE}} = \arg\max_{\boldsymbol{\theta} \in \mathbb{P}} \ell_t(\boldsymbol{\theta})$, (47) holds by the fact that $\log(x+1) \leq x$, and (48) holds by Cauchy–Schwarz inequality under any positive definite matrix $\mathbf{A}$. To handle $\|\boldsymbol{\theta}_{t+1}^{\text{MLE}} - \boldsymbol{\theta}_t^{\text{MLE}}\|_{\mathbf{A}}$, we start from the fact that $\boldsymbol{\theta}_{t+1}^{\text{MLE}} = \arg\max_{\boldsymbol{\theta} \in \mathbb{P}} \ell_{t+1}(\boldsymbol{\theta})$, which leads to the following inequality:

$$0 \leq \ell_{t+1}(\boldsymbol{\theta}_{t+1}^{\text{MLE}}) - \ell_{t+1}(\boldsymbol{\theta}_t^{\text{MLE}}) \tag{49}$$

$$= (\boldsymbol{\theta}_{t+1}^{\text{MLE}} - \boldsymbol{\theta}_t^{\text{MLE}})^\top \nabla_{\boldsymbol{\theta}} \left( \ell_t(\boldsymbol{\theta}) + \log \phi_t(s_{t+1})^\top \boldsymbol{\theta} \right) \big|_{\boldsymbol{\theta} = \boldsymbol{\theta}_t^{\text{MLE}}} - \frac{1}{2} \|\boldsymbol{\theta}_{t+1}^{\text{MLE}} - \boldsymbol{\theta}_t^{\text{MLE}}\|_{\mathbf{A}_{t+1}(\boldsymbol{\theta}_{t+1}')}^2 \tag{50}$$

$$\leq \frac{\phi_t(s_{t+1})^\top}{\phi_t(s_{t+1})^\top \boldsymbol{\theta}_t^{\text{MLE}}} (\boldsymbol{\theta}_{t+1}^{\text{MLE}} - \boldsymbol{\theta}_t^{\text{MLE}}) - \frac{1}{2} \|\boldsymbol{\theta}_{t+1}^{\text{MLE}} - \boldsymbol{\theta}_t^{\text{MLE}}\|_{\mathbf{A}_{t+1}(\boldsymbol{\theta}_{t+1}')}^2, \tag{51}$$

where (50) holds by Taylor's theorem and $\mathbf{A}_{t+1}(\boldsymbol{\theta}_{t+1}') := -\nabla_{\boldsymbol{\theta}}^2 \ell_{t+1}(\boldsymbol{\theta})|_{\boldsymbol{\theta} = \boldsymbol{\theta}_{t+1}'}$ ($\boldsymbol{\theta}_{t+1}'$ is some convex combination between $\boldsymbol{\theta}_{t+1}^{\text{MLE}}$ and $\boldsymbol{\theta}_t^{\text{MLE}}$), and (51) holds due to the necessary condition of optimality for constrained problem as $\nabla_{\boldsymbol{\theta}} \ell_t(\boldsymbol{\theta})|_{\boldsymbol{\theta} = \boldsymbol{\theta}_t^{\text{MLE}}}^\top (\boldsymbol{\theta}_{t+1}^{\text{MLE}} - \boldsymbol{\theta}_t^{\text{MLE}}) \leq 0$. Then, by reordering (51) and applying Cauchy–Schwarz inequality, we have

$$\frac{1}{2} \|\boldsymbol{\theta}_{t+1}^{\text{MLE}} - \boldsymbol{\theta}_t^{\text{MLE}}\|_{\mathbf{A}_{t+1}(\boldsymbol{\theta}_{t+1}')} \leq \left\| \frac{\phi_t(s_{t+1})}{\phi_t(s_{t+1})^\top \boldsymbol{\theta}_t^{\text{MLE}}} \right\|_{\mathbf{A}_{t+1}^{-1}(\boldsymbol{\theta}_{t+1}')}. \tag{52}$$

By plugging $\mathbf{A} = \mathbf{A}_{t+1}(\boldsymbol{\theta}_{t+1}')$ into (48) as well as combining (48) and (52), we have

$$\Delta_{t+1} \leq \Delta_t + 2 \left\| \frac{\phi_t(s_{t+1})}{\phi_t(s_{t+1})^\top \boldsymbol{\theta}_t^{\text{MLE}}} \right\|_{\mathbf{A}_{t+1}^{-1}(\boldsymbol{\theta}_{t+1}')}^2, \tag{53}$$

$$\leq \Delta_t + \frac{2}{p_{\min}^2} \cdot \|\phi_t(s_{t+1})\|_{\mathbf{A}_t^{-1}}^2, \tag{54}$$

where (54) holds by Assumption 1 and $\mathbf{A}_{t+1}^{-1}(\boldsymbol{\theta}_{t+1}') \preceq \mathbf{A}_{t+1}^{-1} \preceq \mathbf{A}_t^{-1}$ given the definition in (13). Then, by applying Lemma 3 to (54), we have

$$\Delta_{t+1} \leq \frac{8d^2}{p_{\min}^2} \log \left( \frac{d\lambda + tL^2}{d} \right), \quad \forall t \in \mathbb{N}. \tag{55}$$

$\square$

# B PROOFS OF THE MAIN THEOREMS

## B.1 PROOF OF THEOREM 1

For ease of exposition, we restate Theorem 1 as follows.

**Theorem.** *At each time $t$, with probability at least $1 - \delta$, we have*

$$\|\boldsymbol{\theta}^* - \boldsymbol{\theta}_t^{\mathrm{MLE}}\|_{\mathbf{A}_t}^2 \le \beta_t, \tag{56}$$

*where $\beta_t := \frac{37d^2}{p_{min}^2} \cdot \log\left(\frac{d\lambda + tL^2}{d}\right) \cdot \log \frac{1}{\delta}$.*

*Proof.* By Corollary 1 and Azuma–Hoeffding inequality in Lemma 5, we have

$$\Pr\left(\ell_{t+1}(\boldsymbol{\theta}_t^{\mathrm{MLE}}) - \ell_{t+1}(\boldsymbol{\theta}^*) - \sum_{i=1}^t z_i \ge \sqrt{2M_t \log \frac{1}{\delta}}\right) \le \delta, \tag{57}$$

where $M_t := \sum_{i=1}^t \left(\log L_{i+1}(\boldsymbol{\theta}_i^{\mathrm{MLE}}) - \log L_i(\boldsymbol{\theta}_{i-1}^{\mathrm{MLE}}) - z_i\right)^2$.

- Regarding $\ell_{t+1}(\boldsymbol{\theta}_t^{\mathrm{MLE}}) - \ell_{t+1}(\boldsymbol{\theta}^*)$, we have

$$\ell_{t+1}(\boldsymbol{\theta}_t^{\mathrm{MLE}}) - \ell_{t+1}(\boldsymbol{\theta}^*) = \ell_t(\boldsymbol{\theta}_t^{\mathrm{MLE}}) - \ell_t(\boldsymbol{\theta}^*) + \log \frac{\phi(s_{t+1}|s_t, a_t)^\top \boldsymbol{\theta}_t^{\mathrm{MLE}}}{\phi(s_{t+1}|s_t, a_t)^\top \boldsymbol{\theta}^*} \tag{58}$$

$$\ge \frac{1}{2}\|\boldsymbol{\theta}_t^{\mathrm{MLE}} - \boldsymbol{\theta}^*\|_{-\nabla_{\boldsymbol{\theta}}^2 \ell_t(\boldsymbol{\theta})|_{\boldsymbol{\theta}=\boldsymbol{\theta}'}}^2 + \log \frac{\phi(s_{t+1}|s_t, a_t)^\top \boldsymbol{\theta}_t^{\mathrm{MLE}}}{\phi(s_{t+1}|s_t, a_t)^\top \boldsymbol{\theta}^*} \tag{59}$$

$$\ge \frac{1}{2}\|\boldsymbol{\theta}_t^{\mathrm{MLE}} - \boldsymbol{\theta}^*\|_{\mathbf{A}_t}^2 - \frac{1}{p_{\min}}, \tag{60}$$

where (59) holds by Taylor's theorem, the necessary condition of optimality for constrained problems $\nabla_{\boldsymbol{\theta}} \ell_t(\boldsymbol{\theta})|_{\boldsymbol{\theta}=\boldsymbol{\theta}_t^{\mathrm{MLE}}}^\top (\boldsymbol{\theta}^* - \boldsymbol{\theta}_t^{\mathrm{MLE}}) \le 0$, and $\boldsymbol{\theta}'$ is some convex combination of $\boldsymbol{\theta}_t^{\mathrm{MLE}}$ and $\boldsymbol{\theta}^*$, and (60) holds by $\mathbf{A}_t \preceq -\nabla_{\boldsymbol{\theta}}^2 \ell_t(\boldsymbol{\theta})|_{\boldsymbol{\theta}=\boldsymbol{\theta}'}$, $\boldsymbol{\theta}^* \in \mathbb{P}$ and Assumption 1.

- For $\sum_{i=1}^t z_i$, denoting $\phi_t(s) := \phi(s|s_t, a_t)$, we have

$$\sum_{i=1}^t z_i = \log\left(\frac{L_t(\boldsymbol{\theta}_t^{\mathrm{MLE}})}{L_t(\boldsymbol{\theta}_{t-1}^{\mathrm{MLE}})} \cdot \frac{L_{t-1}(\boldsymbol{\theta}_{t-1}^{\mathrm{MLE}})}{L_{t-1}(\boldsymbol{\theta}_{t-2}^{\mathrm{MLE}})} \cdot \ldots \cdot \frac{L_1(\boldsymbol{\theta}_1^{\mathrm{MLE}})}{L_1(\boldsymbol{\theta}_0^{\mathrm{MLE}})}\right) \tag{61}$$

$$= \log\left(\frac{\phi_{t-1}(s_t)^\top \boldsymbol{\theta}_t^{\mathrm{MLE}} \cdot \phi_{t-2}(s_{t-1})^\top \boldsymbol{\theta}_t^{\mathrm{MLE}} \cdot \ldots}{\phi_{t-1}(s_t)^\top \boldsymbol{\theta}_{t-1}^{\mathrm{MLE}} \cdot \phi_{t-2}(s_{t-1})^\top \boldsymbol{\theta}_{t-2}^{\mathrm{MLE}} \cdot \ldots}\right) + \frac{\lambda}{2}\left(\|\boldsymbol{\theta}_t^{\mathrm{MLE}}\|_2^2 - \|\boldsymbol{\theta}_0^{\mathrm{MLE}}\|_2^2\right) \tag{62}$$

$$= \sum_{i=1}^{t-1} \log \frac{\phi_i(s_{i+1})^\top \boldsymbol{\theta}_t^{\mathrm{MLE}}}{\phi_i(s_{i+1})^\top \boldsymbol{\theta}_i^{\mathrm{MLE}}} + \frac{\lambda}{2}\left(\|\boldsymbol{\theta}_t^{\mathrm{MLE}}\|_2^2 - \|\boldsymbol{\theta}_0^{\mathrm{MLE}}\|_2^2\right) \tag{63}$$

$$\le \frac{8d^2}{p_{\min}^2} \log\left(\frac{d\lambda + (t-1)L^2}{d}\right), \tag{64}$$

where (64) holds by Lemma 2.

- For $M_t$, we have

$$M_t = \sum_{i=1}^{t} \left( \log L_{i+1}(\boldsymbol{\theta}_i^{\text{MLE}}) - \log L_i(\boldsymbol{\theta}_{i-1}^{\text{MLE}}) - z_i \right)^2 \tag{65}$$

$$= \sum_{i=1}^{t} \left( \log \frac{\phi_i(s_{i+1})^\top \boldsymbol{\theta}_i^{\text{MLE}}}{\phi_i(s_{i+1})^\top \boldsymbol{\theta}^*} \right)^2 \tag{66}$$

$$\leq \frac{1}{p_{\min}^2} \sum_{i=1}^{t} \|\phi_i(s_{i+1})\|_{\mathbf{A}_i^{-1}}^2 \cdot \|\boldsymbol{\theta}_i^{\text{MLE}} - \boldsymbol{\theta}^*\|_{\mathbf{A}_i}^2 \tag{67}$$

$$\leq \frac{\max_{i \leq t} \|\boldsymbol{\theta}_i^{\text{MLE}} - \boldsymbol{\theta}^*\|_{\mathbf{A}_i}^2}{p_{\min}^2} \sum_{i=1}^{t} \|\phi_i(s_{i+1})\|_{\mathbf{A}_i^{-1}}^2 \tag{68}$$

$$\leq \max_{i \leq t} \|\boldsymbol{\theta}_i^{\text{MLE}} - \boldsymbol{\theta}^*\|_{\mathbf{A}_i}^2 \cdot \frac{2d}{p_{\min}^2} \cdot \log\left( \frac{d\lambda + tL^2}{d} \right), \tag{69}$$

where (67) holds by Assumption 1 and Cauchy–Schwarz inequality, and (69) holds by Lemma 3.

Then, combining (60), (64) and (69) into (57), for all $t$, we have

$$\|\boldsymbol{\theta}_t^{\text{MLE}} - \boldsymbol{\theta}^*\|_{\mathbf{A}_t}^2 \leq \frac{1}{p_{\min}} + \frac{8d^2}{p_{\min}^2} \log\left( \frac{d\lambda + tL^2}{d} \right)$$
$$+ \max_{i \leq t} \|\boldsymbol{\theta}_i^{\text{MLE}} - \boldsymbol{\theta}^*\|_{\mathbf{A}_i} \cdot \sqrt{\frac{4d}{p_{\min}^2} \cdot \log\left( \frac{d\lambda + tL^2}{d} \right) \cdot \log \frac{1}{\delta}} \tag{70}$$

holds with probability at least $1 - \delta$. Letting $k_t := \operatorname{argmax}_{i \leq t} \|\boldsymbol{\theta}_i^{\text{MLE}} - \boldsymbol{\theta}^*\|_{\mathbf{A}_i} \leq t$ and denoting the following indicator functions:

$$J := \mathbb{1}\left\{ \|\boldsymbol{\theta}_t^{\text{MLE}} - \boldsymbol{\theta}^*\|_{\mathbf{A}_t}^2 \geq \frac{37d^2}{p_{\min}^2} \cdot \log\left( \frac{d\lambda + tL^2}{d} \right) \cdot \log \frac{1}{\delta} \right\} \tag{71}$$

$$D_i := \mathbb{1}\{k_t = i\}, \forall i \leq t, \tag{72}$$

Then, by (70) and the definition of $k_t$, we have

$$\|\boldsymbol{\theta}_t^{\text{MLE}} - \boldsymbol{\theta}^*\|_{\mathbf{A}_t}^2 \leq \|\boldsymbol{\theta}_{k_t}^{\text{MLE}} - \boldsymbol{\theta}^*\|_{\mathbf{A}_{k_t}}^2 \tag{73}$$

$$\leq \frac{1}{p_{\min}} + \frac{8d^2}{p_{\min}^2} \log\left( \frac{d\lambda + k_t L^2}{d} \right)$$
$$+ \|\boldsymbol{\theta}_{k_t}^{\text{MLE}} - \boldsymbol{\theta}^*\|_{\mathbf{A}_{k_t}} \cdot \sqrt{\frac{4d}{p_{\min}^2} \cdot \log\left( \frac{d\lambda + k_t L^2}{d} \right) \cdot \log \frac{1}{\delta}} \tag{74}$$

holds with probability at least $1 - \delta$, which implies

$$\|\boldsymbol{\theta}_t^{\text{MLE}} - \boldsymbol{\theta}^*\|_{\mathbf{A}_t}^2 \leq \frac{37d^2}{p_{\min}^2} \cdot \log\left( \frac{d\lambda + tL^2}{d} \right) \cdot \log \frac{1}{\delta} := \beta_t. \tag{75}$$

By the fact that $\sum_{i=1}^{t} \Pr(D_i = 1) = 1$ and $\Pr(J|D_i = 1) \leq \delta, \forall i \in [t]$ shown above, we have

$$\Pr(J) = \sum_{i=1}^{t} \Pr(D_i = 1)\Pr(J|D_i = 1) \leq \delta. \tag{76}$$

We complete the proof. $\qquad\square$

**Lemma.** *At each time $t$, with probability at least $1 - \delta$, we have*

$$\nabla_{\boldsymbol{\theta}} \ell_t(\boldsymbol{\theta})|_{\boldsymbol{\theta}=\boldsymbol{\theta}_t^{\text{MLE}}}^\top (\boldsymbol{\theta}_t^{\text{MLE}} - \boldsymbol{\theta}^*) \leq \beta_t', \tag{77}$$

*where $\beta_t' := \frac{22d^2}{p_{min}^2} \log\left( \frac{d\lambda + tL^2}{d} \right) \cdot \max\{1, \log \frac{1}{\delta}\}$.*

*Proof.* This lemma can be proved by the same argument as Theorem 1. By the fact that $\ell_{t+1}(\boldsymbol{\theta}_t^{\text{MLE}}) - \ell_{t+1}(\boldsymbol{\theta}^*) \geq \nabla_{\boldsymbol{\theta}} \ell_t(\boldsymbol{\theta})|_{\boldsymbol{\theta}=\boldsymbol{\theta}_t^{\text{MLE}}}^{\top}(\boldsymbol{\theta}^* - \boldsymbol{\theta}_t^{\text{MLE}}) - \frac{1}{p_{\min}}$, (57), (64), and (69), we also have

$$\nabla_{\boldsymbol{\theta}} \ell_t(\boldsymbol{\theta})|_{\boldsymbol{\theta}=\boldsymbol{\theta}_t^{\text{MLE}}}^{\top}(\boldsymbol{\theta}_t^{\text{MLE}} - \boldsymbol{\theta}^*) \leq \frac{1}{p_{\min}} + \frac{8d^2}{p_{\min}^2} \log\left(\frac{d\lambda + k_t L^2}{d\lambda}\right)$$

$$+ \|\boldsymbol{\theta}_{k_t}^{\text{MLE}} - \boldsymbol{\theta}^*\|_{\mathbf{A}_{k_t}} \cdot \sqrt{\frac{4d}{p_{\min}^2} \cdot \log\left(\frac{d\lambda + k_t L^2}{d}\right) \cdot \log\frac{1}{\delta}} \quad (78)$$

$$\leq \frac{22d^2}{p_{\min}^2} \log\left(\frac{d\lambda + tL^2}{d}\right) \cdot \max\{1, \log\frac{1}{\delta}\} := \beta_t', \quad (79)$$

where (79) holds by plugging (75) into (78). $\qquad\square$

### B.2 PROOF OF THEOREM 2

Recalling that

$$\beta_t := \frac{37d^2}{p_{\min}^2} \cdot \log\left(\frac{d\lambda + tL^2}{d}\right) \cdot \log\frac{1}{\delta} \quad (80)$$

$$\beta_t' = \frac{22d^2}{p_{\min}^2} \log\left(\frac{d\lambda + tL^2}{d}\right) \cdot \max\{1, \log\frac{1}{\delta}\}, \quad (81)$$

we then state the detailed form of the regret upper bound.

**Theorem.** *For all linear kernel MDP $\mathcal{M} = \langle \mathcal{S}, \mathcal{A}, P, R, T, \mu_0 \rangle$, with probability at least $1 - \frac{1}{T} - \delta$, VBMLE, proposed in Algorithm 1, has regret upper bound satisfies that*

$$\mathcal{R}(T) = \sum_{t=1}^{T} (V^*(s_t, \boldsymbol{\theta}^*) - V^{\pi_t}(s_t; \boldsymbol{\theta}^*))$$

$$\leq \left(\frac{\beta_T}{2p_{\min}^2} + \beta_T'\right) \cdot \sum_{t=1}^{T} \frac{1}{\alpha(t)} + \frac{4\gamma}{1-\gamma}\sqrt{T \log\frac{1}{\delta}}$$

$$+ \frac{2\gamma}{p_{\min}(1-\gamma)^2}\sqrt{T \log\frac{1}{\delta}} + \frac{\sqrt{\beta_T}\gamma}{p_{\min}(1-\gamma)^2}\left(1 + \sqrt{T \cdot 2d \log\left(\frac{d\lambda + TL^2}{d}\right)}\right)$$

$$+ \frac{2d \cdot \alpha(T)}{(1-\gamma)} \cdot \log\left(\frac{d\lambda + TL^2}{d}\right). \quad (82)$$

*By choosing $\alpha(t) = \sqrt{t}$, we have $\mathcal{R}(T) = \mathcal{O}(d\sqrt{T} \log T / (p_{\min}^4 (1-\gamma)^2))$.*

*Proof.* By the definition of the cumulative regret in (4), we have

$$\mathcal{R}(T) = \sum_{t=1}^{T} (V^*(s_t, \boldsymbol{\theta}^*) - V^{\pi_t}(s_t; \boldsymbol{\theta}^*)) \quad (83)$$

$$\leq \sum_{t=1}^{T} \left(V^*(s_t, \boldsymbol{\theta}_t^{\mathbf{V}}) - V^{\pi_t}(s_t; \boldsymbol{\theta}^*) + \frac{\ell_t(\boldsymbol{\theta}_t^{\mathbf{V}}) - \ell_t(\boldsymbol{\theta}^*)}{\alpha(t)}\right) \quad (84)$$

$$= \mathcal{R}'(T) + \sum_{t=1}^{T} \frac{\ell_t(\boldsymbol{\theta}_t^{\mathbf{V}}) - \ell_t(\boldsymbol{\theta}^*)}{\alpha(t)} \quad (85)$$

where (84) holds due to the following inequality:

$$\ell_t(\boldsymbol{\theta}_t^{\mathbf{V}}) + \alpha(t) V^*(s_t; \boldsymbol{\theta}_t^{\mathbf{V}}) \geq \ell_t(\boldsymbol{\theta}^*) + \alpha(t) V^*(s_t; \boldsymbol{\theta}^*) \quad (86)$$

$$\implies V^*(s_t; \boldsymbol{\theta}^*) \leq V^*(s_t; \boldsymbol{\theta}_t^{\mathbf{V}}) + \frac{\ell_t(\boldsymbol{\theta}_t^{\mathbf{V}}) - \ell_t(\boldsymbol{\theta}^*)}{\alpha(t)}, \quad (87)$$

and (85) holds by $\mathcal{R}'(T) := \sum_{t=1}^T (V^*(s_t, \boldsymbol{\theta}_{\mathrm{t}}^{\mathbf{V}}) - V^{\pi_t}(s_t; \boldsymbol{\theta}^*))$. For the term $\sum_{t=1}^T (\ell_t(\boldsymbol{\theta}_{\mathrm{t}}^{\mathbf{V}}) - \ell_t(\boldsymbol{\theta}^*))/\alpha(t)$, we have

$$\sum_{t=1}^T \frac{\ell_t(\boldsymbol{\theta}_{\mathrm{t}}^{\mathbf{V}}) - \ell_t(\boldsymbol{\theta}^*)}{\alpha(t)} = \sum_{t=1}^T \frac{\ell_t(\boldsymbol{\theta}_{\mathrm{t}}^{\mathbf{V}}) - \ell_t(\boldsymbol{\theta}_t^{\mathrm{MLE}})}{\alpha(t)} + \sum_{t=1}^T \frac{\ell_t(\boldsymbol{\theta}_t^{\mathrm{MLE}}) - \ell_t(\boldsymbol{\theta}^*)}{\alpha(t)} \tag{88}$$

$$\leq -\sum_{t=1}^T \frac{1}{2\alpha(t)} \|\boldsymbol{\theta}_{\mathrm{t}}^{\mathbf{V}} - \boldsymbol{\theta}_t^{\mathrm{MLE}}\|_{\mathbf{A}_t(\boldsymbol{\theta}')}^2 + \sum_{t=1}^T \frac{1}{2\alpha(t)} \|\boldsymbol{\theta}_t^{\mathrm{MLE}} - \boldsymbol{\theta}^*\|_{\mathbf{A}_t(\boldsymbol{\theta}'')}^2$$
$$-\sum_{t=1}^T \frac{\nabla_{\boldsymbol{\theta}} \ell_t(\boldsymbol{\theta})|_{\boldsymbol{\theta}=\boldsymbol{\theta}_t^{\mathrm{MLE}}}^\top (\boldsymbol{\theta}^* - \boldsymbol{\theta}_t^{\mathrm{MLE}})}{\alpha(t)} \tag{89}$$

$$\leq \sum_{t=1}^T \frac{1}{2p_{\min}^2 \alpha(t)} \|\boldsymbol{\theta}_t^{\mathrm{MLE}} - \boldsymbol{\theta}^*\|_{\mathbf{A}_t}^2 - \sum_{t=1}^T \frac{\nabla_{\boldsymbol{\theta}} \ell_t(\boldsymbol{\theta})|_{\boldsymbol{\theta}=\boldsymbol{\theta}_t^{\mathrm{MLE}}}^\top (\boldsymbol{\theta}^* - \boldsymbol{\theta}_t^{\mathrm{MLE}})}{\alpha(t)}$$
$$-\sum_{t=1}^T \frac{1}{2\alpha(t)} \|\boldsymbol{\theta}_{\mathrm{t}}^{\mathbf{V}} - \boldsymbol{\theta}_t^{\mathrm{MLE}}\|_{\mathbf{A}_t}^2 \tag{90}$$

$$\leq \left(\frac{\beta_T}{2p_{\min}^2} + \beta_T'\right) \cdot \sum_{t=1}^T \frac{1}{\alpha(t)} - \sum_{t=1}^T \frac{1}{2\alpha(t)} \|\boldsymbol{\theta}_{\mathrm{t}}^{\mathbf{V}} - \boldsymbol{\theta}_t^{\mathrm{MLE}}\|_{\mathbf{A}_t(\boldsymbol{\theta}')}^2, \tag{91}$$

where (89) holds by applying Taylor's theorem with $\boldsymbol{\theta}' \in (\boldsymbol{\theta}_{\mathrm{t}}^{\mathbf{V}}, \boldsymbol{\theta}_t^{\mathrm{MLE}}), \boldsymbol{\theta}'' \in (\boldsymbol{\theta}_t^{\mathrm{MLE}}, \boldsymbol{\theta}^*)$, and the fact that $\nabla_{\boldsymbol{\theta}} \ell_t(\boldsymbol{\theta})|_{\boldsymbol{\theta}=\boldsymbol{\theta}_t^{\mathrm{MLE}}}^\top (\boldsymbol{\theta}_{\mathrm{t}}^{\mathbf{V}} - \boldsymbol{\theta}_t^{\mathrm{MLE}}) \leq 0$, (90) holds due to $-p_{\min}^2 \nabla_{\boldsymbol{\theta}}^2 \ell_t(\boldsymbol{\theta}) \preceq \mathbf{A}_t \preceq -\nabla_{\boldsymbol{\theta}}^2 \ell_t(\boldsymbol{\theta}), \forall \boldsymbol{\theta} \in \mathbb{P}$, and (91) holds with probability at least $1 - \sum_{t=1}^T \frac{1}{T^2}$ by (75), (77), and replacing $\delta$ with $\frac{1}{T^2}$ in $\beta_T$ and $\beta_T'$, Lemma 3, and Cauchy–Schwarz inequality. Then, we have

$$\mathcal{R}'(T) = \sum_{t=1}^T \left[ V^*(s_t, \boldsymbol{\theta}_{\mathrm{t}}^{\mathbf{V}}) - V^{\pi_t}(s_t; \boldsymbol{\theta}^*) \right] \tag{92}$$

$$= \gamma \sum_{t=2}^{T+1} \left[ \mathbb{E}_{s' \sim \mathrm{Pr}(\cdot|s_t, a_t; \boldsymbol{\theta}_{\mathrm{t}}^{\mathbf{V}})} [V^*(s', \boldsymbol{\theta}_{\mathrm{t}}^{\mathbf{V}})] - \mathbb{E}_{s' \sim \mathrm{Pr}(\cdot|s_t, a_t; \boldsymbol{\theta}^*)} [V^{\pi_t}(s', \boldsymbol{\theta}^*)] \right] \tag{93}$$

$$= \gamma \sum_{t=2}^{T+1} \Bigg[ \underbrace{\mathbb{E}_{s' \sim \mathrm{Pr}(\cdot|s_t, a_t; \boldsymbol{\theta}^*)} [V^*(s', \boldsymbol{\theta}_{\mathrm{t}}^{\mathbf{V}}) - V^{\pi_t}(s', \boldsymbol{\theta}^*)] - \left( V^*(s_{t+1}, \boldsymbol{\theta}_{\mathrm{t}}^{\mathbf{V}}) - V^{\pi_t}(s_{t+1}, \boldsymbol{\theta}^*) \right)}_{:=B_1}$$
$$+ \underbrace{\mathbb{E}_{s' \sim \mathrm{Pr}(\cdot|s_t, a_t; \boldsymbol{\theta}_{\mathrm{t}}^{\mathbf{V}})} [V^*(s', \boldsymbol{\theta}_{\mathrm{t}}^{\mathbf{V}})] - V^{\pi_t}(s_{t+1}, \boldsymbol{\theta}^*)}_{:=B_2}$$
$$\underbrace{-\mathbb{E}_{s' \sim \mathrm{Pr}(\cdot|s_t, a_t; \boldsymbol{\theta}^*)} [V^*(s', \boldsymbol{\theta}_{\mathrm{t}}^{\mathbf{V}})] + V^*(s_{t+1}, \boldsymbol{\theta}_{\mathrm{t}}^{\mathbf{V}})}_{:=B_3} \Bigg] \tag{94}$$

$$\leq \frac{4\gamma}{1-\gamma} \sqrt{T \log \frac{1}{\delta}} + B_2 \tag{95}$$

where (93) holds by $\pi_t = \arg\max_\pi V^\pi(s_t, \boldsymbol{\theta}_{\mathrm{t}}^{\mathbf{V}})$, and (95) holds with probability at least $1 - \sum_{t=1}^T \frac{2}{t^2}$ by applying Azuma Hoeffding inequality in Lemma 5 on $B_1$ and $B_3$, which are martingale difference

sequences. For $B_2$, we have

$$\gamma \sum_{t=2}^{T+1} \left[ \mathbb{E}_{s' \sim \Pr(\cdot | s_t, a_t; \boldsymbol{\theta}_{\mathrm{t}}^{\mathbf{V}})}[V^*(s', \boldsymbol{\theta}_{\mathrm{t}}^{\mathbf{V}})] - V^{\pi_t}(s_{t+1}, \boldsymbol{\theta}^*) \right]$$

$$= \gamma \sum_{t=2}^{T+1} \left[ \mathbb{E}_{s' \sim \Pr(\cdot | s_t, a_t; \boldsymbol{\theta}^*)} \left[ \frac{\Pr(s' | s_t, a_t; \boldsymbol{\theta}_{\mathrm{t}}^{\mathbf{V}})}{\Pr(s' | s_t, a_t; \boldsymbol{\theta}^*)} V^*(s', \boldsymbol{\theta}_{\mathrm{t}}^{\mathbf{V}}) \right] - V^{\pi_t}(s_{t+1}, \boldsymbol{\theta}^*) \right] \tag{96}$$

$$= \gamma \underbrace{\sum_{t=2}^{T+1} \left[ \mathbb{E}_{s' \sim \Pr(\cdot | s_t, a_t; \boldsymbol{\theta}^*)} \left[ \frac{\Pr(s' | s_t, a_t; \boldsymbol{\theta}_{\mathrm{t}}^{\mathbf{V}})}{\Pr(s' | s_t, a_t; \boldsymbol{\theta}^*)} V^*(s', \boldsymbol{\theta}_{\mathrm{t}}^{\mathbf{V}}) \right] - \frac{\Pr(s_{t+1} | s_t, a_t; \boldsymbol{\theta}_{\mathrm{t}}^{\mathbf{V}})}{\Pr(s_{t+1} | s_t, a_t; \boldsymbol{\theta}^*)} V^*(s_{t+1}, \boldsymbol{\theta}_{\mathrm{t}}^{\mathbf{V}}) \right]}_{:= B_4}$$

$$+ \gamma \sum_{t=2}^{T+1} \left[ \frac{\Pr(s_{t+1} | s_t, a_t; \boldsymbol{\theta}_{\mathrm{t}}^{\mathbf{V}})}{\Pr(s_{t+1} | s_t, a_t; \boldsymbol{\theta}^*)} V^*(s_{t+1}, \boldsymbol{\theta}_{\mathrm{t}}^{\mathbf{V}}) - V^{\pi_t}(s_{t+1}, \boldsymbol{\theta}^*) \right] \tag{97}$$

$$= B_4 + \gamma \mathcal{R}'(T) + \frac{2\gamma}{1-\gamma} + \gamma \sum_{t=2}^{T+1} \left[ \left( \frac{\Pr(s_{t+1} | s_t, a_t; \boldsymbol{\theta}_{\mathrm{t}}^{\mathbf{V}})}{\Pr(s_{t+1} | s_t, a_t; \boldsymbol{\theta}^*)} - 1 \right) V^*(s_{t+1}, \boldsymbol{\theta}_{\mathrm{t}}^{\mathbf{V}}) \right] \tag{98}$$

$$\leq \frac{2\gamma}{p_{\min}(1-\gamma)} \sqrt{T \log \frac{1}{\delta}} + \gamma \underbrace{\sum_{t=2}^{T+1} \left[ \left( \frac{\Pr(s_{t+1} | s_t, a_t; \boldsymbol{\theta}_{\mathrm{t}}^{\mathbf{V}})}{\Pr(s_{t+1} | s_t, a_t; \boldsymbol{\theta}^*)} - 1 \right) V^*(s_{t+1}, \boldsymbol{\theta}_{\mathrm{t}}^{\mathbf{V}}) \right]}_{:= B_5} + \gamma \mathcal{R}'(T) + \frac{2\gamma}{1-\gamma}$$

$$\tag{99}$$

where (96) holds by importance sampling, (97) holds by adding and subtracting $\Pr(s_{t+1} | s_t, a_t; \boldsymbol{\theta}_{\mathrm{t}}^{\mathbf{V}}) / \Pr(s_{t+1} | s_t, a_t; \boldsymbol{\theta}^*) \cdot V^*(s_{t+1}, \boldsymbol{\theta}_{\mathrm{t}}^{\mathbf{V}})$, (98) holds by Lemma 4, and (99) holds with probability at least $1 - \delta$ by applying Azuma-Hoeffding inequality in Lemma 5 on $B_4$. Then, for the term $B_5$, we have

$$B_5 = \gamma \sum_{t=2}^{T+1} \left[ \left( \frac{\Pr(s_{t+1} | s_t, a_t; \boldsymbol{\theta}_{\mathrm{t}}^{\mathbf{V}})}{\Pr(s_{t+1} | s_t, a_t; \boldsymbol{\theta}^*)} - 1 \right) V^*(s_{t+1}, \boldsymbol{\theta}_{\mathrm{t}}^{\mathbf{V}}) \right] \tag{100}$$

$$\leq \frac{\gamma}{p_{\min}(1-\gamma)} \sum_{t=1}^{T} |\langle \phi(s_{t+1} | s_t, a_t), \boldsymbol{\theta}_{\mathrm{t}}^{\mathbf{V}} - \boldsymbol{\theta}^* \rangle| + \frac{\gamma}{p_{\min}(1-\gamma)} \tag{101}$$

$$\leq \frac{\gamma}{p_{\min}(1-\gamma)} \sum_{t=1}^{T} \|\phi(s_{t+1} | s_t, a_t)\|_{\mathbf{A}_t^{-1}} \cdot \|\boldsymbol{\theta}_{\mathrm{t}}^{\mathbf{V}} - \boldsymbol{\theta}_t^{\mathrm{MLE}}\|_{\mathbf{A}_t}$$

$$+ \frac{\gamma}{p_{\min}(1-\gamma)} \sum_{t=1}^{T} \|\phi(s_{t+1} | s_t, a_t)\|_{\mathbf{A}_t^{-1}} \cdot \|\boldsymbol{\theta}_t^{\mathrm{MLE}} - \boldsymbol{\theta}^*\|_{\mathbf{A}_t} + \frac{\gamma}{p_{\min}(1-\gamma)} \tag{102}$$

$$\leq \frac{\gamma}{p_{\min}(1-\gamma)} \sum_{t=1}^{T} \|\phi(s_{t+1} | s_t, a_t)\|_{\mathbf{A}_t^{-1}} \cdot \|\boldsymbol{\theta}_{\mathrm{t}}^{\mathbf{V}} - \boldsymbol{\theta}_t^{\mathrm{MLE}}\|_{\mathbf{A}_t}$$

$$+ \frac{\sqrt{\beta_T} \gamma}{p_{\min}(1-\gamma)} \left( 1 + \sqrt{T \cdot 2d \log \left( \frac{d\lambda + TL^2}{d} \right)} \right) \tag{103}$$

where (101) holds by $p_{\min} \leq \min_{t \leq T} \Pr(s_{t+1} | s_t, a_t; \boldsymbol{\theta}^*)$, which is defined in Assumption 1, (102) holds by Cauchy–Schwarz inequality and triangle inequality, and (103) holds by Theorem 1 with probability at least $1 - \frac{1}{T}$ by replacing $\delta$ with $\frac{1}{T^2}$ in $\beta_T$ and Lemma 3. Combining the final term in

(91) and (103), we have

$$\sum_{t=1}^{T}\left(-\frac{1}{2\alpha(t)}\|\boldsymbol{\theta}_{\mathrm{t}}^{\mathbf{V}}-\boldsymbol{\theta}_t^{\mathrm{MLE}}\|_{\mathbf{A}_t}^2 + \frac{\gamma}{p_{\min}(1-\gamma)}\|\phi(s_{t+1}|s_t,a_t)\|_{\mathbf{A}_t^{-1}}\|\boldsymbol{\theta}_{\mathrm{t}}^{\mathbf{V}}-\boldsymbol{\theta}_t^{\mathrm{MLE}}\|_{\mathbf{A}_t}\right)$$

$$\leq\left(\frac{\gamma}{4p_{\min}(1-\gamma)}\right)^2\sum_{t=1}^{T}\alpha(t)\|\phi(s_{t+1}|s_t,a_t)\|_{\mathbf{A}_t^{-1}}^2 \tag{104}$$

$$=\alpha(T)\cdot 2d\log\left(\frac{d\lambda+TL^2}{d}\right), \tag{105}$$

where (104) holds by completing the square, and (105) holds by Lemma 3. Letting $\alpha(t) = \sqrt{t}$ and combining (85), (91), (95), (99), (103), and (105), we complete the proof. $\qquad\square$

## C   DIFFERENCES BETWEEN VBMLE FOR RL AND VBMLE FOR BANDITS

Compared to the existing works on VBMLE for bandits (Hung et al., 2021; Hung & Hsieh, 2023), VBMLE for RL presents its own salient challenges:

- **Non-Concave Objective Function of VBMLE for Linear Mixture MDPs:** In the context of bandits, VBMLE learns by maximizing the log-likelihood of observed *rewards* with a bias term that depends on the maximum achievable reward. As the parametric form of the reward distributions is typically unknown in the bandit setting, (Hung et al., 2021; Hung & Hsieh, 2023) rely on a surrogate likelihood function (typically belongs to an exponential family) to estimate the unknown reward distributions and incorporates a reward bias by adding the immediate maximum reward. As a result, the resulting objective function still remains a concave function such that its maximizer either enjoys a closed-form expression or could be solved efficiently by a gradient-based method. By contrast, VBMLE for RL manages to optimize the log-likelihood with the value bias, which ends up as a *non-concave* function. To address this issue, we take the following approach: (i) For the theoretical regret analysis, we consider an oracle that returns the maximizer of the constrained optimization problem induced by VBMLE. (ii) For the practical implementation, we could incorporate the value iteration into the optimization subroutine and use a gradient-based method to numerically find an approximate maximizer of VBMLE.

- **Non-Index-Type Policies:** Prior works on VBMLE for bandits (Liu et al., 2020; Hung et al., 2021; Hung & Hsieh, 2023) could convert the original VBMLE into an *index-type policy* by using arm-specific estimators. This conversion also facilitates the regret analysis in (Liu et al., 2020; Hung et al., 2021; Hung & Hsieh, 2023). However, this approach is not applicable in RL for linear mixture MDPs since the action and state spaces could typically be very large in practice. As a result, we are not allowed to reuse a similar analytical framework to characterize the regret of VBMLE in linear mixture MDPs. To address this, we leverage a supermartingale approach and use an induction argument to establish the regret bound, as will be shown in Section 5.

## D   REGRET ANALYSIS FOR ADAPTIVE VBMLE

**Theorem 3.** *For all linear mixture MDP $\mathcal{M} = \langle\mathcal{S}, \mathcal{A}, P, R, T, \mu_0\rangle$, with probability at least $1 - \frac{1}{T} - 3\delta$ and choosing $\alpha(t) = \sqrt{t}$, VBMLE with*

$$\boldsymbol{\theta}_{\mathrm{t}}^{V} := \underset{\boldsymbol{\theta}\in\mathbb{P}_t}{\arg\max}\left\{\sum_{i=1}^{t-1}\log\langle\phi(s_{i+1}|s_i,a_i),\boldsymbol{\theta}\rangle + \frac{\lambda}{2}\|\boldsymbol{\theta}\|_2^2 + \alpha(t)\cdot V^*(s_t;\boldsymbol{\theta})\right\}, \tag{106}$$

*and $\mathbb{P}_t$ is defined in (15), has a regret upper bound that satisfies*

$$\mathcal{R}(T) = \mathcal{O}\left(\frac{d\sqrt{T}(\log T)^5}{(1-\gamma)^2} + \frac{T_0}{1-\gamma}\right), \tag{107}$$

*where $T_0 := \exp(\frac{1}{p_{min}})$.*

*Proof.* The proof closely resembles that of Theorem 1 and Theorem 2 by replacing $p_{\min}$ with $\frac{1}{\log t}$. We highlight the distinctions in Theorem 1 as follows:

- In Lemma 2, the upper bound of $\Delta_t$ is $8d^2(\log t)^2 \log\left(\frac{d\lambda + (t-1)L^2}{d}\right)$ due to $\phi_t(s_{t+1})^\top \boldsymbol{\theta}_t^{\mathrm{MLE}} \geq \frac{1}{\log t}, \forall t \in [T_0, T]$ in (53).

- The final term $\frac{1}{p_{\min}}$ in (60) becomes $\log t, \forall t \in [T_0, T]$.

- For $M_t$, the RHS in (69) becomes $\max_{i \leq t} \|\boldsymbol{\theta}_i^{\mathrm{MLE}} - \boldsymbol{\theta}^*\|_{\mathbf{A}_i}^2 \cdot 2d(\log t)^2 \log\left(\frac{d\lambda + tL^2}{d}\right)$ due to $\phi_i(s_{i+1})^\top \boldsymbol{\theta}^* \geq \frac{1}{\log t}$.

Then, we have

$$\beta_t = 37d^2 \cdot (\log t)^2 \cdot \log\left(\frac{d\lambda + tL^2}{d}\right) \cdot \log\frac{1}{\delta} \tag{108}$$

$$\beta_t' = 22d^2 \cdot (\log t)^2 \cdot \log\left(\frac{d\lambda + tL^2}{d}\right) \cdot \max\{1, \log\frac{1}{\delta}\} \tag{109}$$

We also highlight the distinctions in Theorem 2 as follows:

- The initial steps $T_0$ are chosen such that $\frac{1}{\log T_0} \leq p_{\min}$. The cumulative regret before $T_0$ is equivalent to $\frac{T_0}{1-\gamma}$ according to Lemma 4.

- For the remain cumulative regret, we have $-\frac{1}{(\log t)^2}\nabla_{\boldsymbol{\theta}}^2 \ell_t(\boldsymbol{\theta}) \preceq \mathbf{A}_t \preceq -\nabla_{\boldsymbol{\theta}}^2 \ell_t(\boldsymbol{\theta}), \forall \boldsymbol{\theta} \in \mathbb{P}_t$ applied in (90). Therefore, we have

$$\mathcal{R}(T) = \mathcal{O}\left(\frac{d\sqrt{T}(\log T)^5}{(1-\gamma)^2} + \frac{\exp(\frac{1}{p_{\min}})}{1-\gamma}\right). \tag{110}$$

$\square$

# E  BAYESIAN OPTIMIZATION FOR VBMLE

In this section, we present another approach for addressing the non-concave optimization problem outlined in (10). While we can employ a gradient-based method to identify a $\boldsymbol{\theta}$ that achieves a local maximum in (10), the complexity of such approaches, such as the Trust-Region Constrained Algorithm Byrd et al. (1987) and Coordinate Descent Algorithm Wright (2015), are typically high due to the growth in the number of constraints, which scales in the order of $|\mathcal{S}|^2$. This complexity makes it challenging to tackle large-scale ($|\mathcal{S}| \geq 100$) linear mixture MDPs using VBMLE and other baseline algorithms like Zhou et al. (2021a;b). To find the maximizer of a black-box function $f$ with sample efficiency, Bayesian optimization (BO) leverages the Gaussian process (GP) prior to parameterize each point in the domain $D \subset \mathbb{R}^d$ into a mean function and a covariance function. For choosing a next sample, such BO approaches like EI Močkus (1975), PI Kushner (1964), GP-UCB Srinivas et al. (2010) will maintain a acquisition function (AF) to be an index policy based on the GP estimation. Consequently, we explore the use of GP-UCB to solve $\boldsymbol{\theta}_t^{\mathbf{V}}$. The noisy sample $y_k$ at point $x_k \in D$ satisfies $y_k = f(x_k) + \epsilon_k$, where $\epsilon_k \sim N(0, \sigma^2)$ is Gaussian noise. The regret of BO is defined as $R_K^{\mathrm{BO}} := \sum_{k=1}^K (f(x^*) - f(x_k))$, where $K$ is the total horizon for BO. The following lemma provides the regret bound for GP-UCB Srinivas et al. (2012).

## E.1  REGRET ANALYSIS FOR VBMLE WITH BO

**Lemma 7** (Theorem 3 in Srinivas et al. (2012))**.** If the objective function $f$ lies in the RKHS corresponding to exponential spectral decay kernel $\mathcal{K}(\mathbf{x}, \mathbf{x}')$. Assume that $\|f\|_{\mathcal{K}}^2 \leq B$. The regret for GP-UCB satisfies that

$$\Pr\left\{R_K^{\mathrm{BO}} \leq \sqrt{C_1 H \beta_K \gamma_K} \ \forall K \geq 1\right\} \geq 1 - \delta, \tag{111}$$

where $C_1 = 8/\log(1 + \sigma^{-2})$, $\beta_k = 2B + 300\gamma_k \log^2(k/\delta)$ and $\gamma_k = \mathcal{O}((\log k)^{d+1})$.

Then, we can incorporate the optimization error from BO into the regret bound of VBMLE.

**Theorem 4.** *For all linear mixture MDP $\mathcal{M} = \langle \mathcal{S}, \mathcal{A}, P, R, T, \mu_0 \rangle$, with probability at least $1 - \frac{1}{T} - 4\delta$ and choosing $\alpha(t) = \sqrt{t}$, VBMLE, proposed in Algorithm 1, with parameter selection by GP-UCB has a regret upper bound that satisfies*

$$\mathcal{R}(T) = \mathcal{O}\left( \max\left\{ \frac{d\sqrt{T}\log T}{p_{min^2}(1-\gamma)^2}, \sqrt{T}\frac{(\log K)^{d+1}}{\sqrt{K}} \right\} \right). \tag{112}$$

*Proof.* The proof is almost the same as that in Theorem 2. We highlight the distinctions in Theorem 2 as follows: Let $\boldsymbol{\theta}_t^{\text{BO}}$ to be the maximizer selected by GP-UCB and $E_t^{\text{BO}}$ to be the optimization error at time $t$, which satisfies that

$$\ell_t(\boldsymbol{\theta}_{\text{t}}^{\text{V}}) + \alpha(t)V^*(s_t; \boldsymbol{\theta}_{\text{t}}^{\text{V}}) - (\ell_t(\boldsymbol{\theta}_t^{\text{BO}}) + \alpha(t)V^*(s_t; \boldsymbol{\theta}_t^{\text{BO}})) \leq E_t^{\text{BO}}, \tag{113}$$

and $E_t^{\text{BO}}$ can be handled by Lemma 7. Then, applying the similar property of (86), we have

$$E_t^{\text{BO}} + \ell_t(\boldsymbol{\theta}_t^{\text{BO}}) + \alpha(t)V^*(s_t; \boldsymbol{\theta}_t^{\text{BO}}) \geq \ell_t(\boldsymbol{\theta}^*) + \alpha(t)V^*(s_t; \boldsymbol{\theta}^*) \tag{114}$$

$$\implies V^*(s_t; \boldsymbol{\theta}^*) \leq V^*(s_t; \boldsymbol{\theta}_t^{\text{BO}}) + \frac{\ell_t(\boldsymbol{\theta}_t^{\text{BO}}) - \ell_t(\boldsymbol{\theta}^*)}{\alpha(t)} + \frac{E_t^{\text{BO}}}{\alpha(t)}. \tag{115}$$

Notice that we do not need to handle the optimization error in $\boldsymbol{\theta}_t^{\text{BO}}$ since we apply the technique of completing the square. Simply replace $\boldsymbol{\theta}_{\text{t}}^{\text{V}}$ with $\boldsymbol{\theta}_t^{\text{BO}}$ in (104)-(105). We completed the proof. □

## F  IMPLEMENTATION DETAILS

### F.1  ENVIRONMENT

- $\phi(\cdot|s, a)$: We employ a neural network architecture with two linear hidden layers, each utilizing the Rectified Linear Unit (ReLU) activation function applied to every neuron. The network's input is created by concatenating the one-hot vectors derived from the state and action indices. The output of this network is subsequently transformed into a set of $d$ final layers, each having a dimension of $hidden\_size \times |\mathcal{S}|$, and employing the softmax activation function. The resulting outputs from these final layers are concatenated to yield the final output, which can be represented as $\{\phi(s'|s, a)\}_{s' \in \mathcal{S}} \in \mathbb{R}^{d \times |\mathcal{S}|}$.
- $\boldsymbol{\theta}^*$: By initializing the parameter vector $\boldsymbol{\theta}^*$ with random values such that the summation of its elements equals 1, we can readily verify that the $\boldsymbol{\theta}^*$ resides within the probability simplex.

All the simulations are conducted on the device with (i) CPU: Intel Core i7-11700K, (ii) RAM: 32 GB, (iii) GPU: RTX 3080Ti, and (iv) OS: Windows 10.

### F.2  HYPER-PARAMETERS

| | |
|---|---|
| $\gamma$ | 0.9 |
| Temperature of the softmax function | 0.01 |
| $\lambda$ | 1 |
| $\delta$ for UCLK | 0.1 |
| $d$ | 3 |
| $K$ | 25 |
| length scale for GP | 0.1 |

Figure 3(a) shows the comparison of $\|\boldsymbol{\theta}_{\text{t}}^{\text{V}} - \boldsymbol{\theta}^*\|_2^2$ and $\|\boldsymbol{\theta}_t^{\text{UCLK}} - \boldsymbol{\theta}^*\|_2^2$. Notably, due to UCLK learning distinct parameter for each state-action pair, we also plot $\min_{s,a}\|\boldsymbol{\theta}_t^{\text{UCLK}}(s, a) - \boldsymbol{\theta}^*\|_2^2$ and $\frac{1}{|\mathcal{S}||\mathcal{A}|}\sum_{(s,a)}\|\boldsymbol{\theta}_t^{\text{UCLK}}(s, a) - \boldsymbol{\theta}^*\|_2^2$ for UCLK. The result shows that VBMLE learned a more accurate representation of true parameter $\boldsymbol{\theta}^*$.

# G    ADDITIONAL SIMULATION RESULT

**Empirical regret for the environment with different $p_{\mathbf{min}}$:** As shown in the Theorem 2, the regret bound is depend on the factor of $1/p_{\min}^4$. We highlight that the above dependency on $1/p_{\min}^4$ could be quite conservative in practice. Specifically, we evaluate VBMLE on two MDPs, each with a moderate $p_{\min}$ or a very small $p_{\min}$. The empirical regrets are shown in the Figure 2. The result shows that VBMLE still achieve low empirical regret under a $p_{\min}$ less than $0.001$.

**Distance between the learned $\theta$ and $\theta^*$:** Figure 3 shows the distance between the learned $\theta$ and $\theta^*$ MDP with $|\mathcal{S}| = 5, |\mathcal{A}| = 4$, compared with UCLK, a tabular case algorithm. We don't provide the result of UCLK since it is impractical in terms of computation time. The result includes two variants of the biased term designed in VBMLE:

- **Approximated VBMLE:** The biased term equals to $\sum_{s' \in \mathcal{S}} \langle \phi(s'|s_t, a_t) V^*(s'; \boldsymbol{\theta}_{t-1}^{\mathbf{V}}), \boldsymbol{\theta} \rangle$. Notice that the term $V^*(s'; \boldsymbol{\theta}_{t-1}^{\mathbf{V}})$ is detached from $\boldsymbol{\theta}$.

- **Exact VBMLE:** The biased term $V^*(s_t; \boldsymbol{\theta})$ is constructed by value iteration (2), which is the same implementation as that in Figure 1.

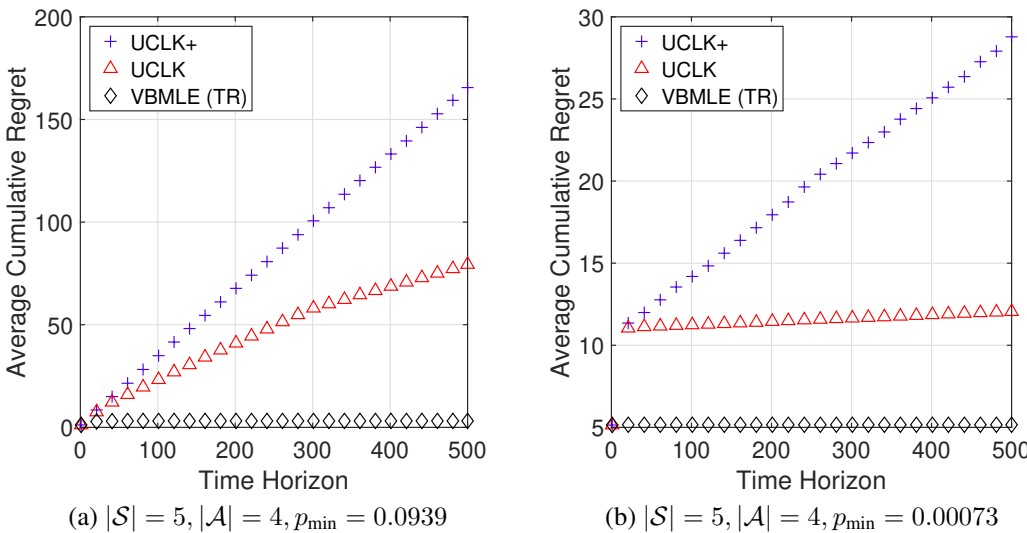

(a) $|\mathcal{S}| = 5, |\mathcal{A}| = 4, p_{\min} = 0.0939$    (b) $|\mathcal{S}| = 5, |\mathcal{A}| = 4, p_{\min} = 0.00073$

Figure 2: Regret averaged over 5 trials.

## G.1    STANDARD DEVIATION OF FIGURE 1

Table 2: The table shows the standard deviation of cumulative regret at $t = 500$.

|  | $|\mathcal{S}| = 3, |\mathcal{A}| = 2$ | $|\mathcal{S}| = 5, |\mathcal{A}| = 4$ |
|---|---|---|
| **UCLK** | 79.289 | 23.982 |
| **VBMLE** | 1.874 | 1.830 |

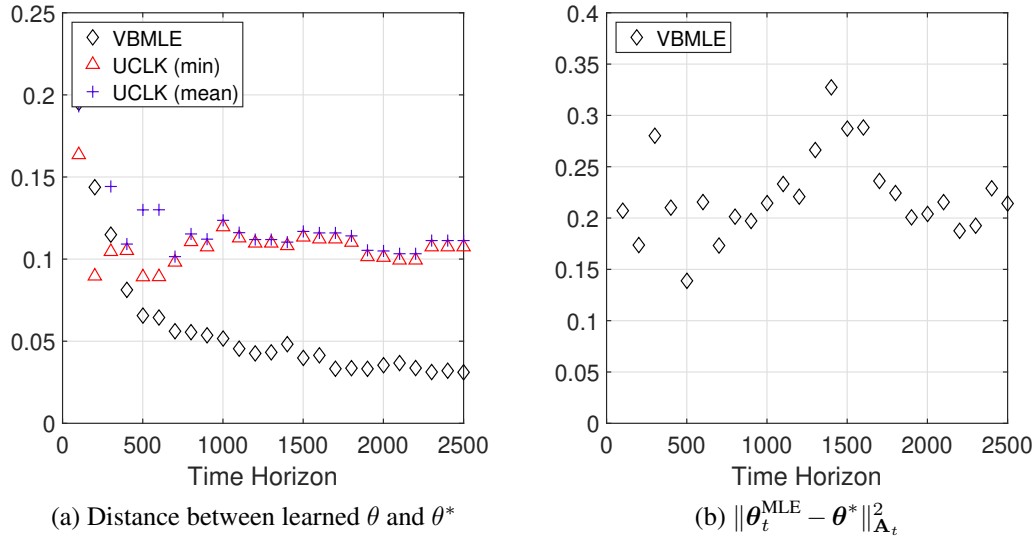

(a) Distance between learned $\theta$ and $\theta^*$      (b) $\|\boldsymbol{\theta}_t^{\text{MLE}} - \boldsymbol{\theta}^*\|_{\mathbf{A}_t}^2$

Figure 3: Observation of the distance over with $T = 2500$.

---

**Algorithm 2** Value Iteration

---

1: **Input:** $\delta, U, \boldsymbol{\theta}$
2: $V^{(1)}(\cdot; \boldsymbol{\theta}) = \frac{1}{1-\gamma}$
3: **for** $u = 1, 2, \cdots, U$ **do**
4:      $V^{(u+1)}(\cdot; \boldsymbol{\theta}) = \max_a \left\{ R(\cdot, a) + \gamma \sum_{s' \in \mathcal{S}} P(s'|s, a; \boldsymbol{\theta}) V^{(u)}(\cdot; \boldsymbol{\theta}) \right\}$
5: **end for**
6: Return $V^{(U+1)}(\cdot; \boldsymbol{\theta})$

---

