# OpenReview forum: "Value-Biased Maximum Likelihood Estimation for Model-based Reinforcement Learning in Discounted Linear MDPs"
_ICLR.cc/2024/Conference — Submitted to ICLR 2024_

### Official Review · Reviewer_WqoR · 2023-10-26

**Soundness:** 3 good
**Presentation:** 3 good
**Contribution:** 2 fair
**Rating:** 5
**Confidence:** 4

**Summary:**

This paper proposes to apply the value-biased maximum likelihood estimation (VBLMLE) principle to solve linear MDP problems. The authors show that the VBMLE method can achieve a $\widetilde{O}(d\sqrt{T})$ regret.  They also demonstrate the proposed method is computationally more efficient than existing methods like UCLK through numerical experiments.

**Strengths:**

* The paper is overall well-written and easy to follow.

* The proposed method is intuitive and well-motivated. The objective for VBMLE clearly exhibits an exploration-exploitation tradeoff.

* The section of theoretical analysis is technically solid, unveiling interesting connections between the proposed methods and the domain of online learning.

**Weaknesses:**

* My main concern is the computational efficiency of the VBMLE method. The authors claim VBMLE method is computationally efficient. But in each time step, VBMLE needs to solve a hard optimization problem (Equation (10) in the paper). And as $t$ gets larger, the problem seems increasingly difficult to handle. The authors say they handle the optimization problem by "incorporating the value iteration into the optimization subroutine". This argument is vague and further explanations are needed. In fact, from the results of numerical experiments, we may find that the VBMLE method needs around 40s to perform one single iteration when solving a small-scale MDP problem ($|\mathcal{S}|=15,|\mathcal{A}|=4$). Although this result is far better than the UCLK algorithm, I feel that the proposed algorithm cannot be called computationally efficient (especially because of its online nature).

* The empirical evaluation is weak. First, the authors only test VBMLE on MDPs with small state/action space. In fact, such tabular problems can be solved without the use of linear function approximation. Second, the authors only compare the proposed method with the UCLK baseline, while there already exists an improved version called $\text{UCLK}^+$ that can achieve near-minimax regret. Also, I think the authors should explain how they solve the inner optimization problem in actual numerical experiments. If there is a problem with page limits, the authors should include the explanations in the Appendix (for example, Appendix C).

* Some notations appear without proper definition. For example, the $\pi_\infty^{\text{MLE}}(s)$ in Section 4.

Given both the strengths and weaknesses of this paper, I choose to give a marginal rating. I suggest the authors should provide more explanations for how they solve the inner-loop optimization problem and try to reduce the computational complexity. (techniques like replay buffer or actor-critic may help?) It would be beneficial to test their algorithm on large-scale MDPs (like what [1] did). I will consider raising my rating if the authors can address my concerns.

[1] Zhang, Tianjun, et al. "Making linear MDPs practical via contrastive representation learning." International Conference on Machine Learning. PMLR, 2022.

**Questions:**

* The ideas behind the VBMLE remind me of model-based RL via the optimistic approach ([2]). Can the authors give a brief discussion about any potential connections between the two approaches?

[2] Jaksch, Thomas, Ronald Ortner, and Peter Auer. "Near-optimal Regret Bounds for Reinforcement Learning." Journal of Machine Learning Research 11 (2010): 1563-1600.

---

> ### Author Response · Authors · 2023-11-23
> **Response to Reviewer WqoR**
>
> ### **Q1: Explain how VBMLE handles the optimization problem in Equation (10)**
>
> The VBMLE maximization problem in (10) can be approximately solved by multiple approaches in practice:
>
> **(i) Solve VBMLE by Bayesian optimization:**
> Due to the non-concavity of VBMLE, we propose to solve VBMLE by Bayesian optimization (BO), which is a powerful and generic method for provably maximizing (possibly non-concave) black-box objective functions. As a result, BO can provably find an $\epsilon$-optimal solution to VBMLE within finite iterations. Specifically:
> - We have applied the GP-UCB algorithm, which is one classic BO algorithm and has been shown to provably find an $\epsilon$-optimal solution within $\tilde{\mathcal{O}}(1/\epsilon^2)$ iterations under smooth (possibly non-concave) objective functions [2]. Each sample taken by GP-UCB requires only one run of standard Value Iteration.
> - To further demonstrate the compatibility of VBMLE and BO, we have extended the regret analysis of VBMLE to the case where only an $\epsilon$-optimal VBMLE solution is obtained (as also suggested by Reviewer p6f5). Specifically, let $K$ denote the number of samples taken by GPUCB in each maximization run of finding VBMLE. We show that VBMLE augmented with BO can achieve a regret bound of  $\mathcal{O}(\max(\frac{d\sqrt{T}\log{T}}{p_{\text{min}^2}(1-\gamma)^2},\sqrt{T}\frac{(\log{K})^{d+1}}{\sqrt{K}} ))$.
> By using a moderate $K$, one could easily recover the same regret bound as that of VBMLE with an exact maximizer. In our experiments, we find that choosing $H=25$ is sufficient and also computationally efficient.
> The empirical regret of VBMLE augmented with BO and other benchmark methods can be found at https://imgur.com/a/8XXHpYJ and in Figure 1 of the updated manuscript.
> The computation times of VBMLE and other methods are provided below https://imgur.com/a/5ksC7Xd and also shown in Table 1 of the updated manuscript. We can see that VBMLE with BO enjoys both low empirical regret as well as low computation time.
>
> We have also added the discussion on VBMLE with BO in Appendix C in the updated manuscript.
>
> **(ii) Approximately solve VBMLE by off-the-shelf constrained optimization solvers:**
> In practice, another possibility is to use an off-the-shelf optimization solver. For example, we have also applied the trust region method (implementation available in SciPy), which is compatible with the VBMLE problem and only requires taking the VBMLE objective function as an input argument (and this is used for constructing the trust-region subproblem, e.g., please refer to [1]). However, as mentioned by the reviewer, one major issue is that due to the non-concavity of VBMLE objective, trust-region methods only ensure convergence to local optima. Despite this, from the experiments, we find that VBMLE augmented with a trust-region method achieves regret comparable to VBMLE with BO.
>
>
> [1] Andrew R. Conn, Nicholas IM Gould, and Philippe L. Toint, “Trust region methods,” Society for Industrial and Applied Mathematics, 2000.
>
> [2] Niranjan Srinivas, Andreas Krause, Sham M Kakade, and Matthias W Seeger, “Information-theoretic regret bounds for Gaussian process optimization in the bandit setting,” IEEE Transactions on Information Theory, 2012.

---

> ### Author Response · Authors · 2023-11-23
> **Response to Reviewer WqoR**
>
> ### **Q2: Compare the computational efficiency of VBMLE and UCLK**
> Thank the reviewer for the helpful suggestion. We provide a detailed comparison as follows:
>
> **1. UCLK suffers from expensive Extended Value Iteration:** In UCLK, the main computational bottleneck lies in the large number of optimization runs needed in Extended Value Iteration (EVI), as shown in the pseudo code of Algorithm 2 in [3] (https://imgur.com/a/J00LX3L). Specifically, there is a quadratic constrained optimization in each Bellman update; Let the number of iterations is denoted by $U$, then each call of EVI will result in solving the quadratic constrained optimization problem for $|S| |A | U$ times. This is rather intractable in practice (e.g., in the experiments, we find that it takes more than 35 hours of wall clock time to finish one single step of UCLK). Similar issues also remain in the improved UCLK+ [4].
>
> **2. VBMLE can efficiently handle the non-concave objective by Bayesian optimization:** Regarding VBMLE, the primary complexity of VBMLE arises from the non-concave constrained optimization in (10), which could be efficiently addressed by using Bayesian optimization (BO) techniques (e.g., GP-UCB), as also described in Q1 and Q2 above. Let $K$ denote the number of samples taken by GPUCB in each maximization run of finding VBMLE (we choose $H=25$ in the experiments). Then, the complexity of VBMLE with GP-UCB for finding $\theta^R_t$ is to solve the standard Value Iteration for only $H+1$ times (H is for BO, each sample requires one value iteration in our objective function, and another 1 for value iteration for $\theta^R_t$). This is a clear computational advantage over the EVI in UCLK.
>
> Moreover, we also conduct experiments to compare the computation times of VBMLE and UCLK, as shown below:
> https://imgur.com/a/5ksC7Xd
> The detailed discussion is also provided in Appendix D in the updated manuscript.
>
> [3] Dongruo Zhou, Jiafan He, and Quanquan Gu, “Provably Efficient Reinforcement Learning for Discounted MDPs with Feature Mapping,” ICML 2021.
>
> [4] Dongruo Zhou, Quanquan Gu, and Csaba Szepesvari, “Nearly Minimax Optimal Reinforcement Learning for Linear Mixture Markov Decision Processes,” COLT 2021.
>
> ### **Q3: Clarify the related work on linear MDPs and linear kernel MDPs**
> In the original manuscript, we follow the terminology used in (Cai et al., 2020), one of the earliest works on this setting, and use the term “linear MDP” for the transition model parameterized as $P(s’|s,a)=<\phi(s’|s,a), \theta^*>$.
> However, we do agree with the reviewer that the term “linear MDP” could be overloaded as “linear MDP” also widely refers to the low-rank factorization where the transition model takes the form of the inner product between a state-action feature vector and the vector of unknown measures over states.
> Therefore, we agree that using the term “linear mixture MDP” could avoid the possible ambiguity and would be more consistent with other existing literature.
>
> We have made the changes accordingly in the updated manuscript. Moreover, for clarity, we also reorganize the Related Work to provide a more thorough discussion on these lines of research works.

---

> ### Author Response · Authors · 2023-11-23
> **Response to Reviewer WqoR**
>
> ### **Q4: Empirical comparison of VBMLE and tabular RL methods as well as UCLK+, and test VBMLE on larger MDPs**
>
> - For a more extensive comparison, we further compare VBMLE (augmented with BO) with Posterior Sampling for Reinforcement Learning (PSRL) [5], a popular benchmark method for tabular RL, on an MDP with |S|=100 and |A|=4. Note that this size is already much larger than those used in the existing linear mixture MDP literature, e.g. Riverswim with |S|=6 in [6] . The results for larger MDPs are provided here: https://imgur.com/a/8XXHpYJ. The results demonstrate that VBMLE converges much faster than PSRL and thereby significantly outperforms PSRL [3] in regret performance, across both small and large MDPs.
>
> - Additionally, we have also incorporated the regret performance of UCLK+ [4] into our experimental results. Note that UCLK+ still suffers from the computational bottleneck of Extended Value Iteration, and hence we could only test UCLK+ on the smaller MDP.
>
> - We also thank the reviewer for providing the helpful reference [7] on the experiments of linear MDPs. We took a careful look at [7] and realized that [7] focuses mainly on linear MDPs and does not handle “linear mixture MDPs” (the difference between two settings are also mentioned in Q3). As “linear mixture MDPs” and linear MDPs are two different settings (and one could not recover the other), it would require a substantially different approach to adapt the technique in [7] to the linear mixture MDP setting. Accordingly, as an alternative, we test VBMLE on the larger MDP with |S|=100 as mentioned above.
>
> [5] Ian Osband, Daniel Russo, and Benjamin Van Roy, “(More) Efficient Reinforcement Learning via Posterior Sampling,” NeurIPS 2013.
>
> [6] Alex Ayoub, Zeyu Jia, Csaba Szepesvari, Mengdi Wang, and Lin F. Yang, “Model-Based Reinforcement Learning with Value-Targeted Regression,” ICML 2020.
>
> [7] Tianjun Zhang et al., "Making linear MDPs practical via contrastive representation learning," ICML 2022.
>
> ### **Q5: Discuss any potential connections between VBMLE and the model-based RL via the optimistic approach (e.g., UCRL-based approaches)**
>
> Indeed, the VBMLE and UCRL approaches are related. UCRL chooses the most optimistic model within a confidence interval, while VBMLE maximizes an objective function which is a linear combination of the value function and the KL-penalty. One can observe that VBMLE optimization problem is a dual formulation of the UCRL optimization (with lagrangian $\frac{1}{\alpha(t)}$). UCRL is a special case of VBMLE for a specific choice of $\alpha(t)$.
> Interestingly, the superior empirical performance of RBMLE suggests that the confidence interval is too big. More detailed discussion on VBMLE vs UCRL can be found in [8] and [9].
>
> [8] Akshay Mete, Rahul Singh, and P. R. Kumar, “Augmented RBMLE-UCB Approach for Adaptive Control of Linear Quadratic Systems,” NeurIPS 2022.
>
> [9] Akshay Mete, Rahul Singh, and P. R. Kumar, "The RBMLE method for Reinforcement Learning," IEEE Annual Conference on Information Sciences and Systems (CISS), 2022.
>
> ### **Q6: Check the typo and notations without definitions (e.g., $\pi_\infty^{\text{MLE}}(s)$ in Section 4)**
>
> Thanks for catching this! We have added the definitions of $\pi_\infty^{\text{MLE}}(s)$ below (7) in our updated manuscript.

---

### Official Review · Reviewer_p6f5 · 2023-10-27

**Soundness:** 3 good
**Presentation:** 1 poor
**Contribution:** 2 fair
**Rating:** 3
**Confidence:** 4

**Summary:**

This paper studies the infinite-horizon linear mixture MDPs with a low-dimensional feature mapping. This paper first proposes a different Value-Biased Maximum Likelihood Estimation based algorithm, which is more computational efficient than previous work. Then this paper provides a theoretical regret bound. In addition, simulation results are provided to validate that VBMLE outperforms previous methods in terms of both empirical regret and computation time.

**Strengths:**

1. This paper considers another algorithm to estimate models and find the optimal policy under linear mixture MDPs with a low-dimensional feature mapping and provides theoretical regret bound.

2. Simulation results are provided to validate theoretical results.

**Weaknesses:**

1. Assumption 1 assumes the lower bound of non-zero transition probabilities to be $p_{min}$. I understand there always exists such a $p_{min}$, but the authors should have discussions on it. I check the proofs that there should be a term like $\frac{1}{p_{min}}$ in regret bound which is hidden by $\mathcal{O}$ notation. However, $1/p_{min}$ can be quite large, even larger than $d$ and $\frac{1}{1-\gamma}$. So in small $p_{min}$ regime, the algorithm is not efficient, and the regret bound result is worse than UCLK [1], which makes the result not good enough. In addition, I suggest to keep $p_{min}$ in the regret bound.

2. This paper claim their better computation efficiency over UCLK [1] because UCLK requires computing an estimate of the model parameter for each state-action pair in each iteration, but it is also hard to solve VBMLE, which is non-convex. Could the authors provide the computation complexity to solve VBMLE, and compare the complexity with the one of UCLK? In addition, for theoretical analysis, the author assume an oracle returning optimal estimators for VBMLE. What if an empircal optimization algorithm is adopt to estimate the parameter? For example, the estimator $\hat{\theta}$ is no longer the optimal one but may have some errors, say $\|\hat{\theta}-\theta^{MLE}\| \leq \epsilon$ for some $\epsilon$.


3. In related work, the linear MDPs in Jin et al., (2020) is different from the model in this paper (which is also called linear mixture MDPs in some literature). I think it will be better if the authors can discuss some other RL works using MLE to estimate transition kernels (For example, FLAMBE [3] and it follow-ups under MDPs and POMDPs) and how the analysis of these work is different from this paper.

Minor

4. I believe the presentation of the this paper can be improved a lot. Many notations and concepts come without definitions and explanations, which makes it hard to follow. Also, there are many typos.

Just to name a few,

Page 3: "All of their algorithm achieve a regret bound of $\mathcal{O}(d\sqrt{H})$", miss $T$; no definition of $\gamma$.

Page 4: no definition of $\pi_{\infty}^{MLE}$.

Page 6:  Eq. (16) "$.... +\sum_{i=1}^{t-1}z_i$". Eq. (21): "$\hat{p}_{min}$".






[1] Dongruo Zhou, Jiafan He, and Quanquan Gu. Provably efficient reinforcement learning for discounted mdps with feature mapping. In International Conference on Machine Learning, pp.
12793–12802. PMLR, 2021

[2] Chi Jin, Zhuoran Yang, Zhaoran Wang, and Michael I Jordan. Provably efficient reinforcement
learning with linear function approximation. In Conference on Learning Theory, pp. 2137–2143.
PMLR, 2020.

[3] Agarwal, A., Kakade, S., Krishnamurthy, A., & Sun, W. (2020). Flambe: Structural complexity and representation learning of low rank mdps. Advances in neural information processing systems, 33, 20095-20107.

**Questions:**

1. In section 6, I am a little confused about " Notably, due to UCLK learning distinct parameter for each state-action pair, ...." . Why is the vector $\theta$ dependent on $(s,a)$?

2. The authors indicate that VBMLE employs this biasing method to handle the exploration-exploitation trade-off. Could the authors explain more in details how to handle the exploration-exploitation trade-off?

---

> ### Author Response · Authors · 2023-11-23
> **Response to Reviewer p6f5**
>
> ### **Q1: Explain how to solve the optimization problem of VBMLE in Equation (10)**
> The VBMLE maximization problem in (10) can be approximately solved by multiple approaches in practice:
>
> **(i) Solve VBMLE by Bayesian optimization:**
> Due to the non-concavity of VBMLE, we propose to solve VBMLE by Bayesian optimization (BO), which is a powerful and generic method for provably maximizing (possibly non-concave) black-box objective functions. As a result, BO can provably find an $\epsilon$-optimal solution to VBMLE within finite iterations. Specifically:
> - We have applied the GP-UCB algorithm, which is one classic BO algorithm and has been shown to provably find an $\epsilon$-optimal solution within $\tilde{\mathcal{O}}(1/\epsilon^2)$ iterations under smooth (possibly non-concave) objective functions [2]. Each sample taken by GP-UCB requires only one run of standard Value Iteration.
> - To further demonstrate the compatibility of VBMLE and BO, we have extended the regret analysis of VBMLE to the case where only an $\epsilon$-optimal VBMLE solution is obtained (as also suggested by Reviewer p6f5). Specifically, let $K$ denote the number of samples taken by GPUCB in each maximization run of finding VBMLE. We show that VBMLE augmented with BO can achieve a regret bound of  $\mathcal{O}(\max(\frac{d\sqrt{T}\log{T}}{p_{\text{min}^2}(1-\gamma)^2},\sqrt{T}\frac{(\log{K})^{d+1}}{\sqrt{K}} ))$.
> By using a moderate $K$, one could easily recover the same regret bound as that of VBMLE with an exact maximizer. In our experiments, we find that choosing $H=25$ is sufficient and also computationally efficient.
> The empirical regret of VBMLE augmented with BO and other benchmark methods can be found at https://imgur.com/a/8XXHpYJ and in Figure 1 of the updated manuscript.
> The computation times of VBMLE and other methods are provided below https://imgur.com/a/5ksC7Xd and also shown in Table 1 of the updated manuscript. We can see that VBMLE with BO enjoys both low empirical regret as well as low computation time.
>
> We have also added the discussion on VBMLE with BO in Appendix C in the updated manuscript.
>
> **(ii) Approximately solve VBMLE by off-the-shelf constrained optimization solvers:**
> In practice, another possibility is to use an off-the-shelf optimization solver. For example, we have also applied the trust region method (implementation available in SciPy), which is compatible with the VBMLE problem and only requires taking the VBMLE objective function as an input argument (and this is used for constructing the trust-region subproblem, e.g., please refer to [1]). However, as mentioned by the reviewer, one major issue is that due to the non-concavity of VBMLE objective, trust-region methods only ensure convergence to local optima. Despite this, from the experiments, we find that VBMLE augmented with a trust-region method achieves regret comparable to VBMLE with BO.
>
>
> [1] Andrew R. Conn, Nicholas IM Gould, and Philippe L. Toint, “Trust region methods,” Society for Industrial and Applied Mathematics, 2000.
>
> [2] Niranjan Srinivas, Andreas Krause, Sham M Kakade, and Matthias W Seeger, “Information-theoretic regret bounds for Gaussian process optimization in the bandit setting,” IEEE Transactions on Information Theory, 2012.

---

> ### Author Response · Authors · 2023-11-23
> **Response to Reviewer p6f5**
>
> ### **Q2: What if an empirical optimization algorithm is adopted to estimate the parameter? For example, the estimator is no longer the optimal one but may have some errors, say $|\hat{\theta}-\theta^{MLE}| \leq \epsilon$ for some $\epsilon$.**
> We thank the reviewer for the insightful suggestion. As mentioned in Q1 above, regarding the empirical optimization algorithm, we propose to apply Bayesian optimization (BO), which can provably find an $\epsilon$-optimal solution of a smooth (possibly non-concave) function within $\tilde{\mathcal{O}}(1/\epsilon^2)$ iterations, to find an approximate solution of VBMLE.
> To further demonstrate the compatibility of VBMLE and BO, we have extended the regret analysis of exact VBMLE to the case where only an $\epsilon$-optimal VBMLE solution is obtained, as suggested by the reviewer. Specifically, let $K$ denote the number of samples taken by GPUCB in each maximization run of finding VBMLE. As described in Theorem 4, we show that VBMLE augmented with BO can achieve a regret bound of  $\mathcal{O}(\max(\frac{d\sqrt{T}\log{T}}{p_{\text{min}^2}(1-\gamma)^2},\sqrt{T}\frac{(\log{K})^{d+1}}{\sqrt{K}} ))$.
> The above regret bound holds for the following reasons: (i) The optimization error of $\theta^{R}$ can be addressed by completing the square in (103), ensuring that even if $\theta^{R}$ is not the exact maximizer, the RHS remains unchanged. (ii) VBMLE does not require the computation of $\theta^{MLE}$ (which only serves as an intermediate machinery for analysis), thus we do not need to take the optimization error for $\theta^{MLE}$ into account. (iii) The optimization error for the "value of the objective function" can be considered and nicely handled by Lemma 7.
> Moreover, this practical variant of VBMLE preserves high computational efficiency and also enjoys low empirical regret, as indicated in the figures below.
> https://imgur.com/a/8XXHpYJ
> https://imgur.com/a/5ksC7Xd
>
>
> ### **Q3: Compare the computational efficiency of VBMLE versus UCLK**
> Thank the reviewer for the helpful suggestion. We provide a detailed comparison as follows:
>
> **1. UCLK suffers from expensive Extended Value Iteration:**
> In UCLK, the main computational bottleneck lies in the large number of optimization runs needed in Extended Value Iteration (EVI), as shown in the pseudo code of Algorithm 2 in [3] (https://imgur.com/a/J00LX3L). Specifically, there is a quadratic constrained optimization in each Bellman update; Let the number of iterations is denoted by $U$, then each call of EVI will result in solving the quadratic constrained optimization problem for $S \times A \times U$ times. This is rather intractable in practice (e.g., in the experiments, we find that it takes more than 35 hours of wall clock time to finish one single step of UCLK). Similar issues also remain in the improved UCLK+ [4].
>
> **2. VBMLE can efficiently handle the non-concave objective by Bayesian optimization:**
> Regarding VBMLE, the primary complexity of VBMLE arises from the non-concave constrained optimization in (10), which could be efficiently addressed by using Bayesian optimization (BO) techniques (e.g., GP-UCB), as also described in Q1 and Q2 above. Let $K$ denote the number of samples taken by GPUCB in each maximization run of finding VBMLE (we choose $H=25$ in the experiments). Then, the complexity of VBMLE with GP-UCB for finding $\theta^R_t$ is to solve the standard Value Iteration for only $H+1$ times (H is for BO, each sample requires one value iteration in our objective function, and another 1 for value iteration for $\theta^R_t$). This is a clear computational advantage over the EVI in UCLK.
>
> Moreover, we also conduct experiments to compare the computation times of VBMLE and UCLK, as shown below:
> https://imgur.com/a/5ksC7Xd
> The detailed discussion is also provided in Appendix D in the updated manuscript.
>
> [3] Dongruo Zhou, Jiafan He, and Quanquan Gu, “Provably Efficient Reinforcement Learning for Discounted MDPs with Feature Mapping,” ICML 2021.
>
> [4] Dongruo Zhou, Quanquan Gu, and Csaba Szepesvari, “Nearly Minimax Optimal Reinforcement Learning for Linear Mixture Markov Decision Processes,” COLT 2021.

---

> ### Author Response · Authors · 2023-11-23
> **Response to Reviewer p6f5**
>
> ### **Q4: Issues with $p_\text{min}$ in VBMLE**
> We would like to clarify the assumption about $p_\text{min}$ and the dependency of the regret bound on $p_{min}$ as follows:
>
> **1. The dependency of the regret bound on the true $p_{min}$**:
> We agree with the reviewer that the regret bound is $p_\text{min}$-dependent and could be shown in the Big-O notation. As shown in the updated Theorem 2, we clearly point out that the regret bound is $\tilde{\mathcal{O}}(\frac{d\sqrt{T}}{p_{\text{min}}^4(1-\gamma)^2})$.
> Moreover, we would like to highlight that the above dependency on $1/p_{min}$ could be quite conservative in practice. Specifically, we evaluate VBMLE on two MDPs, each with a moderate $p_\text{min}$ or a very small $p_{min}$. The empirical regrets are shown in the figures below.
> https://imgur.com/a/Nw4Wp4I
> We could see that VBMLE could still achieve low empirical regret under a $p_\text{min}$ less than 0.001.
>
> We have also added the discussion and the additional experimental results in Appendix F.
>
> **2. The assumption about the knowledge of the true $p_\text{min}$ could be lifted by using a strictly decreasing positive function $p(t)$ with $lim_{t \rightarrow \infty} p(t)=0$ as a surrogate for true $p_\text{min}$:**
> In the original manuscript, we use the knowledge about the true $p_\text{min}$ for two purposes: (i) To ensure that the parameter set $\mathbb{P}$ is not empty; (ii) To provide a lower bound on the term $\phi_t(s_{k+1})^\top \theta_t^{MLE}, \forall k \leq t$.
> Nevertheless, even without the knowledge of the true $p_\text{min}$, the above two requirements could still be met by using a strictly decreasing function $p(t)$ in the parameter set $\mathbb{P}$. Specifically, as $p(t)$ is strictly decreasing and goes to zero in the limit, there must exist some constant $T_0>0$ such that $p(t)< p_{min}$ for all $t>T_0$. As a result, we could remove the assumption about $p_\text{min}$ but with a tradeoff on an additional regret term $T_0$, which is independent of $T$, $d$, and $\gamma$. Under the choice of $p(t)=1/\log(t)$, VBMLE can be shown to achieve $\mathcal{O}\left(\frac{d\sqrt{T}(\log{T})^5}{(1-\gamma)^2}  + \frac{T_0}{1-\gamma}\right)$ regret, without the need for the knowledge of the true $p_\text{min}$.
> For the detailed proof and description, please refer to Theorem 3, Remark 1, and Appendix C.
>
>
> ### **Q5: Clarify the related work on linear MDPs and linear kernel MDPs**
> We thank the reviewer for the suggestion on clarifying the terminology and the related works on these two lines of research.
> In the original manuscript, we follow the terminology used in (Cai et al., 2020), one of the earliest works on this setting, and use the term “linear MDP” for the transition model parameterized as $P(s’|s,a)=<\phi(s’|s,a), \theta^*>$.
> However, we do agree with the reviewer that the term “linear MDP” could be overloaded as “linear MDP” also widely refers to the low-rank factorization where the transition model takes the form of the inner product between a state-action feature vector and the vector of unknown measures over states.
> Therefore, we agree that using the term “linear mixture MDP” could avoid possible ambiguity and would be more consistent with other existing literature.
>
> We have made the changes accordingly in the updated manuscript. Moreover, we also reorganize the Related Work to better highlight the difference between the two types of linear models.
>
>
> ### **Q6: Describe the difference between VBMLE and other RL works on using MLE to estimate transition kernels (e.g., FLAMBE [5])**
> We thank the reviewer for the suggestion and for the reference. The differences between VBMLE and FLAMBE mainly lie in the goal and the problem setting. Specifically,
> - FLAMBE [5] and REP-UCB [6] focus on using MLE for the **representation learning** aspect of low-rank MDPs (i.e., learning the unknown features). As a result, these algorithms and their theoretical guarantees could help enable the downstream RL tasks (e.g., reward maximization or regret minimization), for any given reward function.
>
> - By contrast, VBMLE, as a model-based RL algorithm, uses biased MLE as a subroutine of sequential decision making and focuses on **regret minimization** under linear mixture MDPs, with feature vectors already given. Therefore, VBMLE needs to address the exploration-exploitation issue in solving the linear mixture MDPs (while FLAMBE does not).
>
> Based on the above, we can see that VBMLE and FLAMBE / REP-UCB focus on different aspects of RL under function approximation and therefore could potentially complement each other.
>
> [5] A. Agarwal, S. Kakade, A. Krishnamurthy, and W. Sun, “FLAMBE: Structural complexity and representation learning of low rank MDPs,” NeurIPS 2020.
>
> [6] Masatoshi Uehara, Xuezhou Zhang, and Wen Sun, “Representation learning for online and offline RL in low-rank MDPs,” ICLR 2021.

---

> ### Author Response · Authors · 2023-11-23
> **Response to Reviewer p6f5**
>
> ### **Q7: Explain why vector $\theta$ in UCLK is dependent on (s,a)?**
> As also described in Q3 above, UCLK involves Extended Value Iteration (EVI), and there is a quadratic constrained optimization in each Bellman update (please also refer to the pseudo code of EVI in UCLK: https://imgur.com/a/J00LX3L).
> As a result, if the number of iterations needed in each call of EVI is denoted by $U$, then each EVI shall result in solving the quadratic optimization problem for $|S| |A| U$ times.
>
>
> ### **Q8: Explain how VBMLE handles exploration-exploitation trade-off**
> The bias term in VBMLE controls the exploration and the part of MLE controls the exploitation. If we have $\alpha(t) = 0$, the VBMLE serves as MLE to be a greedy policy based on current observation. In VBMLE, we show that the choice $\alpha(t)=\sqrt{t}$ leads to an optimal trade-off between exploration and exploitation (cf. the statement of Theorem 2 in Appendix B).
>
> ### **Q9: Check the typo and notations without definitions**
> Thanks for catching this. We have fixed the typos and added the definitions of $\pi_\infty^{\text{MLE}}(s)$ below (7) in our updated manuscript.

---

### Official Review · Reviewer_Jxj8 · 2023-11-02

**Soundness:** 3 good
**Presentation:** 3 good
**Contribution:** 3 good
**Rating:** 6
**Confidence:** 3

**Summary:**

The primary focus of this paper revolves around the learning of discounted linear Mixture Markov Decision Processes (MDPs). Diverging from previous frameworks, the author introduces the VBMLE algorithm, which directly learns the parameterized transition without the need for a value-target regression process. This innovative algorithm attains a regret bound of $O(d\sqrt{T}/(1-\gamma)^2)$ with reduced computational complexity. Additionally, the empirical results provide strong evidence confirming the effectiveness of the proposed algorithm.

**Strengths:**

1. The VBMLE algorithm eliminates the value-target regression process and achieves a regret bound of $O(d\sqrt{T}/(1-\gamma)^2)$, resulting in reduced computational complexity.

2. The empirical results offer compelling evidence that affirms the effectiveness of the proposed algorithm.

3. Exploring the connection between online learning and the analysis of regret can be an independent and valuable area of interest.

**Weaknesses:**

1. There exist some error in the related works. The linear MDP setting is first introduced in the Jin et al., 2020 [1], which assume the transition probability satisfies $P(s'|s,a)=\langle  \theta(s,a),\mu(s')\rangle$. In comparision, this work focuse on the linear mixture MDPs setting that $P(s'|s,a)=\langle  \phi(s'|s,a),\theta\rangle$, which is first introduced in Ayoub et al.,2020 [2] and Zhou et al., 2021[3]. Providing accurate references and explanations for these concepts is essential to avoid any confusion or misinterpretation.

[1] Provably efficient reinforcement learning with linear function approximation.

[2] Model-based reinforcement learning with value-targeted regression

[3] Provably efficient reinforcement learning for discounted mdps with feature mapping

2.In contrast to previous work, this study relies on prior knowledge of the zero transition set $P_0$ and assumes a lower bound for non-zero transition probabilities. While these assumptions are made to support the VBMLE algorithm, it's important to recognize that they may restrict the broader applicability of the algorithm.

3. Regarding computational complexity, solving the optimization problem outlined in equation (10) may still pose challenges in terms of efficiency.

**Questions:**

1. The standard Azuma-Hoeffding Inequality requires $|X_t-X_{t-1}|\leq c_t$ and set $M_t=\sum_{i=1}^t c_t^2$. In Lemma 3, it appears that the Azuma-Hoeffding Inequality is being extended to handle adaptive $c_t = X_t - X_{t-1}$, which is indeed a notable extension. Further clarification or explanation from the author on this extension would be beneficial.

2. In the experimental section, it is necessary to explicitly state the chosen dimensions $d$. Additionally, the experiments are conducted on a relatively small state-action space. Under this situation, tabular method may also have good performance. Conducting experiments with larger state-action spaces or comparing the results with tabular methods would indeed provide valuable insights into the algorithm's scalability and efficiency.

3. The author misses some related work about the linear mixture MDP. For instance, He et al., 2021 [1] utilized Bernstein-type concentration properties to improve results in Cai et al., 2020 [2] and attained a near-optimal regret guarantee. Including these references would enhance the completeness of the related works section.

[1] Near-optimal Policy Optimization Algorithms for Learning Adversarial Linear Mixture MDPs

[2]  Provably efficient exploration in policy optimization.

---

> ### Author Response · Authors · 2023-11-23
> **Response to Reviewer Jxj8**
>
> ### **Q1: Solving the optimization problem of VBMLE in Equation (10)**
> The VBMLE maximization problem in (10) can be approximately solved by multiple approaches in practice:
>
> **(i) Solve VBMLE by Bayesian optimization:**
> Due to the non-concavity of VBMLE, we propose to solve VBMLE by Bayesian optimization (BO), which is a powerful and generic method for provably maximizing (possibly non-concave) black-box objective functions. As a result, BO can provably find an $\epsilon$-optimal solution to VBMLE within finite iterations. Specifically:
> - We have applied the GP-UCB algorithm, which is one classic BO algorithm and has been shown to provably find an $\epsilon$-optimal solution within $\tilde{\mathcal{O}}(1/\epsilon^2)$ iterations under smooth (possibly non-concave) objective functions [2]. Each sample taken by GP-UCB requires only one run of standard Value Iteration.
> - To further demonstrate the compatibility of VBMLE and BO, we have extended the regret analysis of VBMLE to the case where only an $\epsilon$-optimal VBMLE solution is obtained (as also suggested by Reviewer p6f5). Specifically, let $K$ denote the number of samples taken by GPUCB in each maximization run of finding VBMLE. We show that VBMLE augmented with BO can achieve a regret bound of  $\mathcal{O}(\max(\frac{d\sqrt{T}\log{T}}{p_{\text{min}^2}(1-\gamma)^2},\sqrt{T}\frac{(\log{K})^{d+1}}{\sqrt{K}} ))$.
> By using a moderate $K$, one could easily recover the same regret bound as that of VBMLE with an exact maximizer. In our experiments, we find that choosing $H=25$ is sufficient and also computationally efficient.
> The empirical regret of VBMLE augmented with BO and other benchmark methods can be found at https://imgur.com/a/8XXHpYJ and in Figure 1 of the updated manuscript.
> The computation times of VBMLE and other methods are provided below https://imgur.com/a/5ksC7Xd and also shown in Table 1 of the updated manuscript. We can see that VBMLE with BO enjoys both low empirical regret as well as low computation time.
>
> We have also added the discussion on VBMLE with BO in Appendix C in the updated manuscript.
>
> **(ii) Approximately solve VBMLE by off-the-shelf constrained optimization solvers:**
> In practice, another possibility is to use an off-the-shelf optimization solver. For example, we have also applied the trust region method (implementation available in SciPy), which is compatible with the VBMLE problem and only requires taking the VBMLE objective function as an input argument (and this is used for constructing the trust-region subproblem, e.g., please refer to [1]). However, as mentioned by the reviewer, one major issue is that due to the non-concavity of VBMLE objective, trust-region methods only ensure convergence to local optima. Despite this, from the experiments, we find that VBMLE augmented with a trust-region method achieves regret comparable to VBMLE with BO.
>
>
> [1] Andrew R. Conn, Nicholas IM Gould, and Philippe L. Toint, “Trust region methods,” Society for Industrial and Applied Mathematics, 2000.
>
> [2] Niranjan Srinivas, Andreas Krause, Sham M Kakade, and Matthias W Seeger, “Information-theoretic regret bounds for Gaussian process optimization in the bandit setting,” IEEE Transactions on Information Theory, 2012.

---

> ### Author Response · Authors · 2023-11-23
> **Response to Reviewer Jxj8**
>
> ### **Q2: Issues with $p_{min}$ in VBMLE**
> We would like to clarify the assumption about $p_{min}$ and the dependency of the regret bound on $p_{min}$ as follows:
>
> **1. The dependency of the regret bound on the true $p_{min}$:**
> We agree with the reviewer that the regret bound is $p_\text{min}$-dependent and could be shown in the Big-O notation. As shown in the updated Theorem 2, we clearly point out that the regret bound is $\tilde{\mathcal{O}}(\frac{d\sqrt{T}}{p_{\text{min}}^4(1-\gamma)^2})$.
> Moreover, we would like to highlight that the above dependency on $1/p_{min}$ could be quite conservative in practice. Specifically, we evaluate VBMLE on two MDPs, each with a moderate $p_\text{min}$ or a very small $p_{min}$. The empirical regrets are shown in the figures below.
> https://imgur.com/a/Nw4Wp4I
> We could see that VBMLE could still achieve low empirical regret under a $p_\text{min}$ less than 0.001.
>
> We have also added the above discussion and the additional experimental results in Appendix F.
>
> **2. The assumption about the knowledge of the true $p_\text{min}$ could be lifted by using a strictly decreasing positive function $p(t)$ with $lim_{t \rightarrow \infty} p(t)=0$ as a surrogate for true $p_\text{min}$:**
> In the original manuscript, we use the knowledge about the true $p_\text{min}$ for two purposes: (i) To ensure that the parameter set $\mathbb{P}$ is not empty; (ii) To provide a lower bound on the term $\phi_t(s_{k+1})^\top \theta_t^{MLE}, \forall k \leq t$.
> Nevertheless, even without the knowledge of the true $p_\text{min}$, the above two requirements could still be met by using a strictly decreasing function $p(t)$ in the parameter set $\mathbb{P}$. Specifically, as $p(t)$ is strictly decreasing and goes to zero in the limit, there must exist some constant $T_0>0$ such that $p(t)< p_{min}$ for all $t>T_0$. As a result, we could remove the assumption about $p_\text{min}$ but with a tradeoff on an additional regret term $T_0$, which is independent of $T$, $d$, and $\gamma$. Under the choice of $p(t)=1/log(t)$, VBMLE can be shown to achieve $\mathcal{O}\left(\frac{d\sqrt{T}(\log{T})^5}{(1-\gamma)^2}  + \frac{T_0}{1-\gamma}\right)$ regret, without the knowledge of the true $p_\text{min}$.
> For the detailed proof and description, please refer to Theorem 3, Remark 1, and Appendix C.
>
> ### **Q3: Clarify the related work on linear MDPs and linear kernel MDPs**
> We thank the reviewer for the helpful suggestions on clarifying the terminology and for bringing some additional references to our attention.
> In the original manuscript, we follow the terminology used in (Cai et al., 2020), one of the earliest works on this setting, and use the term “linear MDP” for the transition model parameterized as $P(s’|s,a)=<\phi(s’|s,a), \theta^*>$.
> However, we do agree with the reviewer that the term “linear MDP” could be overloaded as “linear MDP” also widely refers to the low-rank factorization where the transition model takes the form of the inner product between a state-action feature vector and the vector of unknown measures over states.
> Therefore, we agree that using the term “linear mixture MDP” could avoid the possible ambiguity and would be more consistent with other existing literature.
>
> We have made the changes accordingly in the updated manuscript. Moreover, for clarity, we also reorganize the Related Work to provide a more thorough discussion on these lines of research works.
>
> ### **Q4: Comparison of VBMLE and tabular RL methods and experiments on larger MDPs**
> We thank the reviewer for the helpful suggestion. For a more extensive evaluation, we further compare VBMLE (augmented with BO) with Posterior Sampling for Reinforcement Learning (PSRL) [3], a popular benchmark method for tabular RL, on an MDP with |S|=100 and |A|=4. Note that this size is already much larger than those used in the existing linear mixture MDP literature, e.g. Riverswim with |S|=6 in [4] . The results for larger MDPs are provided here: https://imgur.com/a/8XXHpYJ. The results demonstrate that VBMLE converges much faster than PSRL and thereby significantly outperforms PSRL [3] in regret performance, across both small and large MDPs.
>
> [3] Ian Osband, Daniel Russo, and Benjamin Van Roy, “(More) Efficient Reinforcement Learning via Posterior Sampling,” NeurIPS 2013.
>
> [4] Alex Ayoub, Zeyu Jia, Csaba Szepesvari, Mengdi Wang, and Lin F. Yang, “Model-Based Reinforcement Learning with Value-Targeted Regression,” ICML 2020.

---

### Official Review · Reviewer_W2xc · 2023-11-02

**Soundness:** 3 good
**Presentation:** 3 good
**Contribution:** 3 good
**Rating:** 3
**Confidence:** 5

**Summary:**

This paper studies reinforcement learning (RL) on MDP with linear decomposition. The authors proposed a MLE-based algorithm to directly estimate the MDP, unlike previous approaches which need to solve a regression problem on the estimated value functions and need to construct a confidence set of the parameters of the MDP. They also provided experiments to show the superiority of the proposed algorithm compared with existing baselines, especially on the computational time aspect.

**Strengths:**

The intuition of the proposed approach is very clear. The authors successfully showed the strengths of the proposed algorithm from both the theory side and experiment side. The proof is also technically sound.

**Weaknesses:**

My main concern is the significance of this work given the works about RL with general function approximation. The main novelty of this work is an MLE-based approach with an additional optimal value function bias term. However, similar approaches have already been considered and analyzed in previous works about RL with posterior sampling [1,2]. In detail, [2] analyzed RL with general function approximation, and the proposed algorithm (Algorithm 3 in [2]) is nearly the same as the algorithm proposed by the authors. The only difference is due to the problem setting (e.g., discounted v.s. episodic, frequentist vs bayesian). Given the fact that the MDP instance considered in this work is just a special case of the MDP instances considered in [2], I can not find enough contribution or novelty of this work.

[1] Zhang, Tong. "Feel-good thompson sampling for contextual bandits and reinforcement learning." SIAM Journal on Mathematics of Data Science 4.2 (2022): 834-857.

[2] Zhong, Han, et al. "Gec: A unified framework for interactive decision making in mdp, pomdp, and beyond." arXiv preprint arXiv:2211.01962 (2022).

Meanwhile, the presentation of this paper is no clear enough, especially for some key points of the theoretical claims and experiment implementations. For example,

1. The big-O notation in this work is not well-defined. In Theorem 2, the authors used a big-O notation to propose the regret of their proposed algorithm, which only depends on terms $T, \gamma$ and $d$. However, from their proof (e.g., (97)) which is used to bound the regret) the regret should also depends on $1/p_{min}$, which is the smallest transition probability $p(s'|s,a)$. The first issue is that, $1/p_{min}$ could be very large, which could make the regret bound derived in Theorem 2 very loose. The second issue is that, the regret bound of UCLK does not depend on $1/p_{min}$, and the authors did not mention or compare them. Therefore, I suggest the authors to revise their big-O notation definition and give a detailed comparison of regret between their algorithm and UCLK.

2. The term 'linear MDP' used in this paper, which refers to MDP instance whose transition probability can be decomposed by a feature $\phi(s'|s,a)$, is not correct enough, as previous works [3,4] claimed such an instance as 'linear kernel MDP' or 'linear mixture MDP'.

[3] Dongruo Zhou, Jiafan He, and Quanquan Gu. Provably efficient reinforcement learning for discounted mdps with feature mapping. In International Conference on Machine Learning, pp.
12793–12802. PMLR, 2021b.

[4] Dongruo Zhou, Quanquan Gu, and Csaba Szepesvari. Nearly minimax optimal reinforcement learning for linear mixture markov decision processes. In Conference on Learning Theory, pp. 4532–
4576. PMLR, 2021a.

3. There also lack some experiment implementation details. From the description of the proposed algorithm (at (14)), the parameter $\theta$ should be selected from a set $\mathbb{P}$, which requires the parameter $p_{min}$. The authors did not mention how to find or estimate such a quantity in their experiment.

**Questions:**

See Weaknesses section.

---

> ### Author Response · Authors · 2023-11-23
> **Response to Reviewer W2xc**
>
> ### **Q1: Explain the implementation of VBMLE in practice**
> We would like to highlight that the VBMLE maximization problem in (10) can be approximately solved by multiple approaches in practice:
>
> **(i) Solve VBMLE by Bayesian optimization:**
> Due to the non-concavity of VBMLE, we propose to solve VBMLE by Bayesian optimization (BO), which is a powerful and generic method for provably maximizing (possibly non-concave) black-box objective functions. As a result, BO can provably find an $\epsilon$-optimal solution to VBMLE within finite iterations. Specifically:
> - We have applied the GP-UCB algorithm, which is one classic BO algorithm and has been shown to provably find an $\epsilon$-optimal solution within $\tilde{\mathcal{O}}(1/\epsilon^2)$ iterations under smooth (possibly non-concave) objective functions [2]. Each sample taken by GP-UCB requires only one run of standard Value Iteration.
> - To further demonstrate the compatibility of VBMLE and BO, we have extended the regret analysis of VBMLE to the case where only an $\epsilon$-optimal VBMLE solution is obtained (as also suggested by Reviewer p6f5). Specifically, let $K$ denote the number of samples taken by GPUCB in each maximization run of finding VBMLE. We show that VBMLE augmented with BO can achieve a regret bound of  $\mathcal{O}\(\max(\frac{d\sqrt{T}\log{T}}{p_{\text{min}^2}(1-\gamma)^2},\sqrt{T}\frac{(\log{K})^{d+1}}{\sqrt{K}} ))$.
> By using a moderate $K$, one could easily recover the same regret bound as that of VBMLE with an exact maximizer. In our experiments, we find that choosing $H=25$ is sufficient and also computationally efficient.
> The empirical regret of VBMLE augmented with BO and other benchmark methods can be found at https://imgur.com/a/8XXHpYJ and in Figure 1 of the updated manuscript.
> The computation times of VBMLE and other methods are provided below https://imgur.com/a/5ksC7Xd and also shown in Table 1 of the updated manuscript. We can see that VBMLE with BO enjoys both low empirical regret as well as low computation time.
>
> We have also added the discussion on VBMLE with BO in Appendix C in the updated manuscript.
>
> **(ii) Approximately solve VBMLE by off-the-shelf constrained optimization solvers:**
> In practice, another possibility is to use an off-the-shelf optimization solver. For example, we have also applied the trust region method (implementation available in SciPy), which is compatible with the VBMLE problem and only requires taking the VBMLE objective function as an input argument (and this is used for constructing the trust-region subproblem, e.g., please refer to [1]). However, as mentioned by the reviewer, one major issue is that due to the non-concavity of VBMLE objective, trust-region methods only ensure convergence to local optima. Despite this, from the experiments, we find that VBMLE augmented with a trust-region method achieves regret comparable to VBMLE with BO.
>
> [1] Andrew R. Conn, Nicholas IM Gould, and Philippe L. Toint, “Trust region methods,” Society for Industrial and Applied Mathematics, 2000.
>
> [2] Niranjan Srinivas, Andreas Krause, Sham M Kakade, and Matthias W Seeger, “Information-theoretic regret bounds for Gaussian process optimization in the bandit setting,” IEEE Transactions on Information Theory, 2012.

---

> ### Author Response · Authors · 2023-11-23
> **Response to Reviewer W2xc**
>
> ### **Q2: Clarify the novelty of VBMLE**
>
> As highlighted by the reviewer, VBMLE and GPS-IDM [3] differ in two key aspects, and we would like to argue that **these two aspects are actually fundamental, require totally different regret analysis, and lead to difference in efficiency:**
>
> **1. Infinite-horizon vs Episodic (and hence the novel techniques involved in regret analysis):**
> The problem settings are different, given that GPS-IDM focuses on episodic MDPs, while VBMLE is tailored to solving infinite-horizon MDPs. Without the reset capability of episodic MDPs, infinite-horizon discounted MDPs pose a unique challenge of tackling planning and exploration in one single trajectory. Indeed, in many applications, it is practically impossible to reset the state of the system. This difference has also been recognized by the existing literature (e.g., [4] and [5]).
>
> In such a non-episodic setting, the regret analysis is more involved and needs to better handle the mixing behavior of the MDP. Specifically, to address this salient challenge of non-episodic setting, we introduce a new convergence result of MLE in Theorem 1 by (i) constructing a novel submartingale and (ii) connecting the linear mixture MDPs with Follow-the-Leader algorithm in online learning. These two techniques indeed handle the mixing behavior of infinite-horizon MDPs.
>
> **2. Computational advantage of VBMLE (due to its frequentist style):**
> We agree with the reviewer that GPS-IDM can be viewed as the Bayesian counterpart of VBMLE (as they both use the value function for exploration). That being said, by taking the frequentist viewpoint, VBMLE could enjoy a computational advantage over its Bayesian counterpart. Specifically:
> - In GPS-IDM, the update of posterior distribution can be computationally expensive in practice. As shown in Line 3 of Algorithm 3 in [3], **each posterior update requires doing one run of standard Value Iteration for every possible model in the hypothesis class**. Note that the hypothesis class could be fairly large in practice, especially under the $\epsilon$-bracket number technique for discretizing the continuous domain of model parameters (cf Appendix G in [3]), and hence the computational complexity could be substantial.
>
> - By contrast, VBMLE takes the frequentist perspective and only requires a small number of runs of Value Iteration when it is augmented with BO (as described in Q1). This manifests the computational efficiency of VBMLE.
>
> [3] Han Zhong et al., "GEC: A Unified Framework for Interactive Decision Making in MDP, POMDP, and beyond," arXiv preprint arXiv:2211.01962, 2022.
>
> [4] Yuanzhou Chen, Jiafan He, and Quanquan Gu, “On the Sample Complexity of Learning Infinite-horizon Discounted Linear Kernel MDPs,” ICML 2022.
>
> [5] Dongruo Zhou, Jiafan He, and Quanquan Gu, “Provably Efficient Reinforcement Learning for Discounted MDPs with Feature Mapping,” ICML 2021.

---

> ### Author Response · Authors · 2023-11-23
> **Response to Reviewer W2xc**
>
> ### **Q3: Issues with $p_{\text{min}}$ in VBMLE**
> We would like to clarify the assumption about $p_{\text{min}}$ and the dependency of the regret bound on $p_{\text{min}}$ as follows:
>
> **1. The dependency of the regret bound on the true $p_{\text{min}}$:**
> We agree with the reviewer that the regret bound is $p_\text{min}$-dependent and could be shown in the Big-O notation. As shown in the updated Theorem 2, we clearly point out that the regret bound is $\tilde{\mathcal{O}}(\frac{d\sqrt{T}}{p_{\text{min}}^4(1-\gamma)^2})$.
> Moreover, we would like to highlight that the above dependency on $1/p_{min}$ could be quite conservative in practice. Specifically, we evaluate VBMLE on two MDPs, each with a moderate $p_\text{min}=0.0939$ and a very small $p_{\text{min}}=0.00073$, respectively. The empirical regrets are shown in the figures below.
> https://imgur.com/a/Nw4Wp4I
>
> We could see that VBMLE could still achieve low empirical regret under a $p_\text{min}$ less than 0.001.
>
> We have also added the above discussion and the additional experimental results in Appendix F.
>
> **2. The assumption about the knowledge of the true $p_\text{min}$ could be lifted by using a strictly decreasing positive function $p(t)$ with $\lim_{t \rightarrow \infty} p(t)=0$ as a surrogate for true $p_\text{min}$:**
> In the original manuscript, we use the knowledge about the true $p_\text{min}$ for two purposes: (i) To ensure that the parameter set $\mathbb{P}$ is not empty; (ii) To provide a lower bound on the term $\phi_t(s_{k+1})^\top \theta_t^{MLE}, \forall k \leq t$.
> Nevertheless, even without the knowledge of the true $p_\text{min}$, the above two requirements could still be met by using a strictly decreasing function $p(t)$ in the parameter set $\mathbb{P}$. Specifically, as $p(t)$ is strictly decreasing and goes to zero in the limit, there must exist some constant $T_0>0$ such that $p(t)< p_{\text{min}}$ for all $t>T_0$. As a result, we could remove the assumption about $p_\text{min}$ but with a tradeoff on an additional regret term $T_0$, which is independent of $T$, $d$, and $\gamma$. Under the choice of $p(t)=1/log(t)$, VBMLE can be shown to achieve $\mathcal{O}\left(\frac{d\sqrt{T}(\log{T})^5}{(1-\gamma)^2}  + \frac{T_0}{1-\gamma}\right)$ regret, without the knowledge of the true $p_\text{min}$.
> For the detailed proof and description, please refer to Theorem 3, Remark 1, and Appendix C.
>
> ### **Q4: The terminology of linear MDPs and linear mixture MDPs**
> We thank the reviewer for the suggestion on clarifying the terminology.
> In the original manuscript, we follow the terminology used in (Cai et al., 2020), one of the earliest works on this setting, and use the term “linear MDP” for the transition model parameterized as $P(s’|s,a)=<\phi(s’|s,a), \theta^*>$.
> However, we do agree with the reviewer that the term “linear MDP” could be overloaded as “linear MDP” also widely refers to the low-rank factorization where the transition model takes the form of the inner product between a state-action feature vector and the vector of unknown measures over states.
> Therefore, we agree that using the term “linear mixture MDP” could avoid the possible ambiguity and would be more consistent with other existing literature. We have made the changes accordingly in the updated manuscript.
>
> ### **Q5: Clarify the experimental and implementation details**
> - Regarding the parameter set $\mathbb{P}$: In the experiments, to showcase that VBMLE is competitive without the need for $p_\text{min}$, we opted not to utilize the parameter set defined in (14) for solving VBMLE; instead, we applied the standard simplex constraints, i.e., directly replacing $p_\text{min}$ with 0 in (14). Notably, the empirical results in Figure 1 in the original manuscript demonstrate that VBMLE indeed remains competitive without the knowledge of the true $p_\text{min}$.
> - Additionally, we conduct another experiment in an environment with a relatively small $p_{\text{min}}$, and the results https://imgur.com/a/Nw4Wp4I indicate satisfactory regret performance, even without applying the knowledge of known $p_{\text{min}}$ in Assumption 1 to VBMLE.
>
> For clarity and better reproducibility, we also add the above description and the experimental results to Appendix E in the updated manuscript.

---

### Meta-Review · Area_Chair_o8po · 2023-12-07

**Metareview:**

The authors provide a computationally efficient algorithm for linear mixture MDP and present some empirical evaluation.

The contribution focuses in the linear setting, but a reviewer notices that such algorithm is actually a special case of a more general procedure with function approximation.
Moreover, the reviewers have additional concerns, such as the empirical evaluation being very week (small MDPs).
A key element in the discussion is about the actual computational efficiency of the algorithm, which questions the actual contribution of the paper. Other issues include the presence of the minimum probability p_mim.

**Justification For Why Not Higher Score:**

The paper seems to make a contribution that is questionable in their claim computational efficiency. Several other issues (see above) also exist.

**Justification For Why Not Lower Score:**

N/A

---

### Decision · Program_Chairs · 2024-01-16

Reject